# VECTOR QUANTIZED WASSERSTEIN AUTO-ENCODER

## ABSTRACT

Learning deep discrete latent presentations offers a promise of better symbolic and summarized abstractions that are more useful to subsequent downstream tasks. Recent work on Vector Quantized Variational Auto-Encoder (VQ-VAE) has made substantial progress in this direction. However, this quantizes latent representations using the online k-means algorithm which suffers from poor initialization and non-stationary clusters. To strengthen the clustering quality for the latent representations, we propose Vector Quantized Wasserstein Auto-Encoder (VQ-WAE) intuitively developed based on the clustering viewpoint of Wasserstein (WS) distance. Specifically, we endow a discrete distribution over the codewords and learn a deterministic decoder that transports the codeword distribution to the data distribution via minimizing a WS distance between them. We develop further theories to connect it with the clustering viewpoint of WS distance, allowing us to have a better and more controllable clustering solution. Finally, we empirically evaluate our method on several well-known benchmarks, where it achieves better qualitative and quantitative performances than the baselines in terms of the codebook utilization and image reconstruction/generation.

## 1 INTRODUCTION

Learning compact yet expressive representations from large-scale and high-dimensional unlabeled data is an important and long-standing task in machine learning (Kingma & Welling, 2013; Chen et al., 2020; Chen & He, 2021; Zoph et al., 2020). Among many different kinds of methods, Variational Auto-Encoder (VAE) (Kingma & Welling, 2013) and its variants (Tolstikhin et al., 2017; Alemi et al., 2016; Higgins et al., 2016; Voloshynovskiy et al., 2019) have shown great success in unsupervised representation learning. Although these continuous representation learning methods have been applied successfully to various problems ranging from images (Pathak et al., 2016; Goodfellow et al., 2014; Kingma et al., 2016), video and audio (Reed et al., 2017; Oord et al., 2016; Kalchbrenner et al., 2017), in some contexts, input data are more naturally modeled and encoded as discrete symbols rather than continuous ones. For example, discrete representations are a natural fit for complex reasoning, planning and predictive learning (Van Den Oord et al., 2017). This motivates the need of learning discrete representations, preserving the insightful characteristics of input data.

Vector Quantization Variational Auto-Encoder (VQ-VAE) (Van Den Oord et al., 2017) is a pioneer generative model, which successfully combines the VAE framework with discrete latent representations. In particular, the vector quantized models learn a compact discrete representation using a deterministic encoder-decoder architecture in the first stage, and subsequently applied this highly compressed representation for various downstream tasks, examples including image generation (Esser et al., 2021), cross-modal translation (Kim et al., 2022), and image recognition (Yu et al., 2021). While VQ-VAE has been widely applied to representation learning in many areas (Henter et al., 2018; Baevski et al., 2020; Razavi et al., 2019; Kumar et al., 2019; Dieleman et al., 2018; Yan et al., 2021), it is known to suffer from *codebook collapse*, which has a low codebook usage, *i.e.* most of embedded latent vectors are quantized to just few discrete codewords, while the *other codewords are rarely used, or dead*, due to the poor initialization of the codebook, reducing the information capacity of the bottleneck (Roy et al., 2018; Takida et al., 2022; Yu et al., 2021).

To mitigate this issue, additional training heuristics were proposed, such as the exponential moving average (EMA) update (Van Den Oord et al., 2017; Razavi et al., 2019), soft expectation maximization (EM) update (Roy et al., 2018), codebook reset (Dhariwal et al., 2020; Williams et al., 2020). Notably, soft expectation maximization (EM) update (Roy et al., 2018) connects the EMA update

with an EM algorithm and softens the EM algorithm with a stochastic posterior. *Codebook reset* randomly reinitializes unused/low-used codewords to one of the encoder outputs (Dhariwal et al., 2020) or those near codewords of high usage Williams et al. (2020). Takida et al. (2022) suspects that deterministic quantization is the cause of codebook collapse and extends the standard VAE with stochastic quantization and trainable posterior categorical distribution, showing that the annealing of the stochasticity of the quantization process significantly improves the codebook utilization.

Additionally, WS distance has been applied successfully to *generative models* and *continuous representation learning* (Arjovsky et al., 2017; Gulrajani et al., 2017; Tolstikhin et al., 2017) owing to its nice properties and rich theory. It is natural to ask: *"Can we take advantages of intuitive properties of the WS distance and its mature theory for learning highly compact yet expressive discrete representations?"* Toward this question, in this paper, we develop solid theories by connecting the theory bodies and viewpoints of the WS distance, generative models, and deep discrete representation learning. In particular, **a)** we first endow a discrete distribution over the codebook and propose learning a "deterministic decoder transporting the codeword to data distributions" via minimizing the WS distance between them; **b)** To devise a trainable algorithm, we develop Theorem 3.1 to equivalently turn the above WS minimization to push-forwarding the data to codeword distributions via minimizing a WS distance between "the latent representation and codeword distributions"; **c)** More interestingly, our Corollary 3.1 proves that when minimizing the WS distance between the latent representation and codeword distributions, the codewords tend to flexibly move to the clustering centroids of the latent representations with a control on the proportion of latent representations associated to a centroid. We argue and empirically demonstrate that using the clustering viewpoint of a WS distance to learn the codewords, we can obtain more *controllable* and *better centroids* than using a simple k-means as in VQ-VAE (cf. Sections 3.1 and 5.2).

Our method, called *Vector Quantized Wasserstein Auto-Encoder* (VQ-WAE), applies the WS distance to learn a more controllable codebook, hence leading to an improvement in the codebook utilization. We conduct comprehensive experiments to demonstrate our key contributions by comparing with VQ-VAE (Van Den Oord et al., 2017) and SQ-VAE (Takida et al., 2022) (*i.e.*, the recent work that can improve the codebook utilization). The experimental results show that our VQ-WAE can achieve better codebook utilization with higher codebook perplexity, hence leading to lower (compared with VQ-VAE) or comparable (compared with SQ-VAE) reconstruction error, with significantly lower reconstructed Fréchlet Inception Distance (FID) score (Heusel et al., 2017). Generally, a better quantizer in the stage-1 can naturally contribute to stage-2 downstream tasks (Yu et al., 2021; Zheng et al., 2022). To further demonstrate this, we conduct comprehensive experiments on four benchmark datasets for both unconditional and class-conditional generation tasks. The experimental results indicate that from the codebooks of our VQ-WAE, we can generate better images with lower FID scores.

## 2 VECTOR QUANTIZED VARIATIONAL AUTO-ENCODER

Given a training set $\mathbb{D} = \{x_1, ..., x_N\} \subset \mathbb{R}^V$, VQ-VAE (Van Den Oord et al., 2017) aims at learning *a codebook which is formed by set of codewords* $C = [c_k]_{k=1}^K \in \mathbb{R}^{K \times D}$ on the latent space $\mathcal{Z} \in \mathbb{R}^D$, *an encoder* $f_e$ to map the data examples to the codewords, and *a decoder* $f_d$ (i.e., $q(x \mid z)$) to reconstruct accurately the data examples from the codewords. Given a data example $x$, the encoder $f_e$ (i.e., $p(z \mid x)$) associates $x$ to the codeword $\bar{f}_e(x) = c$ defined as

$$c = \text{argmin}_k d_z(f_e(x), c_k),$$

where $d_z$ is a metric on the latent space.

The objective function of VQ-VAE is as follows:

$$\mathbb{E}_{x \sim \mathbb{P}_x} \left[ d_x \left( f_d \left( \bar{f}_e(x) \right), x \right) + d_z \left( \textbf{sg} \left( f_e(x) \right), C \right) + \beta d_z \left( f_e(x), \textbf{sg}(C) \right) \right],$$

where $\mathbb{P}_x = \frac{1}{N} \sum_{n=1}^N \delta_{x_n}$ is the empirical data distribution, **sg** specifies stop gradient, $d_x$ is a cost metric on the data space, and $\beta$ is set between 0.1 and 2.0 (Van Den Oord et al., 2017) and $d_z(f_e(x), C) = \sum_{c \in C} d_z(f_e(x), c)$.

The purpose of VQ-VAE training is to form the latent representations in clusters and adjust the codewords to be the centroids of these clusters.

# 3 CONTROLLABLE CODEBOOKS WITH WASSERSTEIN QUANTIZATION

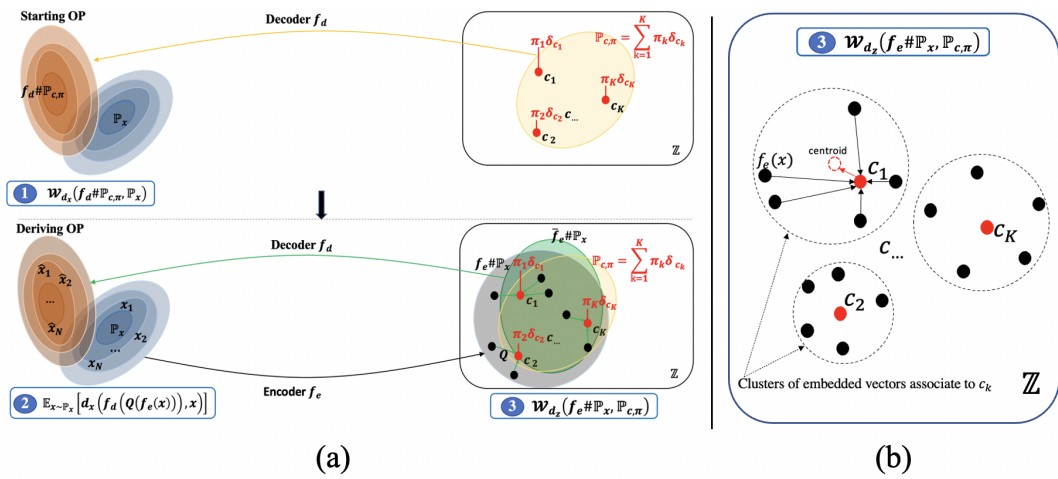

(a)                                                    (b)

Figure 1: (a): Illustration of our VQ-WAE derivation. We depart with the minimization of the WS distance on the data space in (1) and further turn it to minimizing the reconstruction error in (2) and the WS distance on the latent space in (3); (b): Visualisation of the embedding space with WS regularization. The output of the encoder $f_e(x)$ is distributed and moved to codewords $c_k$ in which the cardinalities $\left|\sigma^{-1}(k)\right|$ (i.e., the number of latent representation which are assigned to $k^{th}$ codeword) are proportional to $\pi_k$. At the same time, the codewords tend to flexibly move to the clustering centroids of the latent representations (cf. Corollary 3.1).

We present the theoretical development of our VQ-WAE framework which connects the viewpoints of the WS distance, generative models, and deep discrete representation learning in Section 3.1. Specifically, we propose to learn a *"deterministic decoder transporting the codeword to data distributions"* via minimizing the WS distance between them (Figure 1a (Top)). We then turn the above WS minimization to push-forwarding the data to codeword distribution via minimizing a WS distance between *"the latent representation and codeword distributions"* (Figure 1a (Bottom)). We prove that when minimizing the WS distance between the latent representation and codeword distributions, the codewords tend to flexibly move to the clustering centroids of the latent representations with a control on the proportion of latent representations associated with a centroid (Figure 1b). Based on the theoretical development, we devise a practical algorithm for VQ-WAE in Section 3.2.

## 3.1 THEORETICAL DEVELOPMENT

Given a training set $\mathbb{D} = \{x_1, ..., x_N\} \subset \mathbb{R}^V$, we wish to learn a *codebook* $C = \{c_k\}_{k=1}^{K} \subset \mathbb{R}^{K \times D}$ on a latent space $\mathcal{Z}$ and *an encoder* to map each data example to a given codebook, preserving insightful characteristics carried in the data. We first endow a discrete distribution over the codewords as $\mathbb{P}_{c,\pi} = \sum_{k=1}^{K} \pi_k \delta_{c_k}$ with the Dirac delta function $\delta$ and the weights $\pi \in \Delta_{K-1} = \{\pi' \geq \mathbf{0} : \|\pi'\|_1 = 1\}$.

We aim to learn a decoder function $f_d : \mathcal{Z} \to \mathcal{X}$ (i.e., mapping from the latent space $\mathcal{Z} \subset \mathbb{R}^D$ to the data space $\mathcal{X}$), the codebook $C$, and the weights $\pi$, to minimize:

$$\min_{C,\pi} \min_{f^d} \mathcal{W}_{d_x} \left( f_d \# \mathbb{P}_{c,\pi}, \mathbb{P}_x \right), \tag{1}$$

where $\mathbb{P}_x = \frac{1}{N} \sum_{n=1}^{N} \delta_{x_n}$ is the empirical data distribution and $d_x$ is a cost metric on the data space.

We interpret the optimization problem (OP) in Eq. (1) as follows. Given a discrete distribution $\mathbb{P}_{c,\pi}$ on the codewords, we use the decoder $f_d$ to map the codebook $C$ to the data space and consider $\mathcal{W}_{d_x} \left( f_d \# \mathbb{P}_c, \mathbb{P}_x \right)$ as the *codebook-data distortion* w.r.t. $f_d$. We subsequently learn $f_d$ to minimize the codebook-data distortion given $\mathbb{P}_{c,\pi}$ and finally adjust the codebook $C$ and $\pi$ to minimize the optimal codebook-data distortion. To offer more intuition for the OP in Eq. (1), we introduce the following lemma.

**Lemma 3.1.** *Let $C^* = \{c_k^*\}_k, \pi^*$, and $f_d^*$ be the optimal solution of the OP in Eq. (1). Assume $K < N$, then $C^* = \{c_k^*\}_k, \pi^*$, and $f_d^*$ are also the optimal solution of the following OP:*

$$\min_{f_d} \min_{\pi} \min_{\sigma \in \Sigma_\pi} \sum_{n=1}^{N} d_x \left( x_n, f_d \left( c_{\sigma(n)} \right) \right), \tag{2}$$

*where $\Sigma_\pi$ is the set of assignment functions $\sigma : \{1, ..., N\} \to \{1, ..., K\}$ such that the cardinalities $\left| \sigma^{-1}(k) \right|, k = 1, ..., K$ are proportional to $\pi_k, k = 1, ..., K$.[1]*

Lemma 3.1 states that for the optimal solution $C^* = \{c_k^*\}, \pi^*$, and $f_d^*$ of the OP in Eq. (1), $\{f_d^*(c_k^*)\}_{k=1}^{K}$ become the optimal clustering centroids of the optimal clustering solution which minimizes the distortion. Inspired by Wasserstein Auto-Encoder (Tolstikhin et al., 2017), we establish the following theorem to engage the OP in (1) with the latent space.

**Theorem 3.1.** *We can equivalently turn the optimization problem in (1) to*

$$\min_{C, \pi, f_d} \min_{\bar{f}_e : \bar{f}_e \# \mathbb{P}_x = \mathbb{P}_{c, \pi}} \mathbb{E}_{x \sim \mathbb{P}_x} \left[ d_x \left( f_d \left( \bar{f}_e(x) \right), x \right) \right], \tag{3}$$

*where $\bar{f}_e$ is a **deterministic discrete** encoder mapping data example $x$ directly to the codebook.*

First, we learn both the codebook $C$ and the weights $\pi$. Second, ours seeks a *deterministic discrete* encoder $\bar{f}_e$ mapping data example $x$ directly to a codeword, concurring with vector quantization and serving our further derivations, whereas Theorem 1 in Tolstikhin et al. (2017) involves a *probabilistic/stochastic* encoder mapping to a continuous latent distribution (i.e., a larger space to search). More importantly, our proof is totally different from that in Tolstikhin et al. (2017) (all proof details are given in Appendix A).

Additionally, $\bar{f}_e$ is a deterministic discrete encoder mapping a data example $x$ directly to a codeword. To make it trainable, we replace $\bar{f}_e$ by a continuous encoder $f_e : \mathcal{X} \to \mathcal{Z}$ and arrive the OP:

$$\min_{C, \pi} \min_{f_d, f_e} \left\{ \mathbb{E}_{x \sim \mathbb{P}_x} \left[ d_x \left( f_d \left( Q_C \left( f_e(x) \right) \right), x \right) \right] + \lambda \mathcal{W}_{d_z} \left( f_e \# \mathbb{P}_x, \mathbb{P}_{c, \pi} \right) \right\}, \tag{4}$$

where $Q_C(f_e(x)) = \operatorname{argmin}_{c \in C} d_z(f_e(x), c)$ is a *quantization operator* which returns the closest codeword to $f_e(x)$ and the parameter $\lambda > 0$.

Particularly, we can rigorously prove that the two optimization problems of interest in (3) and (4) are equivalent under some mild conditions in Theorem 3.2. This rationally explains why we could solve the OP in (4) for our final tractable solution.

**Theorem 3.2.** *If we seek $f_d$ and $f_e$ in a family with infinite capacity (e.g., the family of all measurable functions), the three OPs of interest in (1, 3, and 4) are equivalent.*

Moreover, the OP in (4) conveys important meaningful interpretations. Specifically, by minimizing $\mathcal{W}_{d_z}(f_e \# \mathbb{P}_x, \mathbb{P}_{c, \pi})$ w.r.t. $C, \pi$, we aim to learn the codewords that are clustering centroids of $f_e \# \mathbb{P}_x$ according to the clustering viewpoint of OT as shown in Corollary 3.1, and similar to VQ-VAE, we quantize $f_e(x)$ to the closest codeword using $Q_C(f_e(x)) = \operatorname{argmin}_{c \in C} d_z(f_e(x), c)$ and try to reconstruct $x$ from this codebook.

**Corollary 3.1.** *Consider minimizing the second term: $\min_{f_e, C} \mathcal{W}_{d_z}(f_e \# \mathbb{P}_x, \mathbb{P}_{c, \pi})$ in (4) given $\pi$ and assume $K < N$, its optimal solution $f_e^*$ and $C^*$ are also the optimal solution of the OP:*

$$\min_{f_e, C} \min_{\sigma \in \Sigma_\pi} \sum_{n=1}^{N} d_z \left( f_e(x_n), c_{\sigma(n)} \right), \tag{5}$$

*where $\Sigma_\pi$ is the set of assignment functions $\sigma : \{1, ..., N\} \to \{1, ..., K\}$ such that the cardinalities $\left| \sigma^{-1}(k) \right|, k = 1, ..., K$ are proportional to $\pi_k, k = 1, ..., K$.*

Corollary 3.1 indicates the aim of minimizing the second term $\mathcal{W}_{d_z}(f_e \# \mathbb{P}_x, \mathbb{P}_{c, \pi})$ in (4). By which, we adjust the encoder $f_e$ and the codebook $C$ such that the codewords of $C$ become the clustering

---

[1] E.g., $\sigma$ is the nearest assignment: $\sigma^{-1}(k) = \{\bar{f}_e(x) = c_k \mid k = \operatorname{argmin}_k d_z(f_e(x), c_k)\}$ is set of latent representations which are quantized to $k^{th}$ codeword.

centroids of the latent representations $\{f_e(x_n)\}_n$ to minimize the *codebook-latent distortion* (see Figure 1 (Right)). Additionally, at the optimal solution, the optimal assignment function $\sigma^*$, which indicates how latent representations (or data examples) associated with the clustering centroids (i.e., the codewords) has a valuable property, i.e., *the cardinalities* $\left|(\sigma^*)^{-1}(k)\right|, k = 1, ..., K$ *are proportional to* $\pi_k, k = 1, ..., K$.

**Remark:** Recall the codebook collapse issue, i.e. most of embedded latent vectors are quantized to just few discrete codewords while the other codewords are rarely used. Corollary 3.1 give us important properties: *(1) we can control the number of latent representations assigned to each codeword by adjust $\pi$, guaranteeing all codewords are utilized, (2) codewords become the clustering centroids of the associated latent representations to minimize the codebook-latent distortion*, to develop our VQ-WAE framework.

## 3.2 PROPOSED FRAMEWORK

One of crucial aims of learning meaningful and well-distributed codewords is to make use of each individual codeword efficiently by solving the OP in (4). Specifically, we wish the latent representations are more uniformly associated with the codewords. Based on Corollary 3.1, pointing out that *the numbers of latent representations* associated with *the $k^{th}$ codeword* is proportional to $\pi_k$, we hence fix $\pi$ as a uniform distribution (i.e., $\mathbb{P}_{c,\pi} = \sum_{k=1}^{K} \frac{1}{K} \delta_{c_k}$) to make all the codewords utilized equally by the model, hence boosting the perplexity or the codebook usage.

We now present the practical method based on the OP in (4) with $\mathbb{P}_{c,\pi} = \sum_{k=1}^{K} \frac{1}{K} \delta_{c_k}$. At each iteration, we sample a mini-batch $x_1, ..., x_B$ and then solve the OP in (4) by updating $f_d, f_e$ and $C$ based on this mini-batch as follows. Let us denote $\mathbb{P}_b = \frac{1}{B} \sum_{j=i}^{B} \delta_{x_i}$ as the empirical distribution of embedded vectors. over the current batch. Basically, we learn the optimal transportation plan $P^*$ by solving:

$$\mathcal{W}_{d_z}\left(f_e \# \mathbb{P}_b, \mathbb{P}_{c,\pi}\right) = \min_{P \in \Gamma(1_B, 1_C)} \langle P, D_{c,x} \rangle, \tag{6}$$

where $1_B = \left[\frac{1}{B}\right]_B$ is the vector of atom masses of $\mathbb{P}_b$, $1_C = \left[\frac{1}{C}\right]_C$ is the vector of atom masses of $\mathbb{P}_{c,\pi}$, $\Gamma(1_B, 1_C)$ is the set of feasible transportation plans, and $D_{c,x} = [d_z(x_i, c_k)]_{i,k} \in \mathbb{R}^{B \times K}$ is the cost matrix.

The pseudcode of our VQ-WAE is summarized in Algorithm 1. We use the copy gradient trick (Van Den Oord et al., 2017) to deal with the back-propagation from decoder to encoder for reconstruction term while Wasserstein regularization term $\mathcal{W}_{d_z}(f_e \# \mathbb{P}_b, \mathbb{P}_{c,\pi})$ can be optimized directly without further manipulation. Additionally, $\mathcal{W}_{d_z}(f_e \# \mathbb{P}_b, \mathbb{P}_{c,\pi})$ term is only utilized in the training phase.

---

**Algorithm 1** VQ-WAE

---

1: **Initialize**: encoder $f_e$, decoder $f_d$ and codebook $C$.
2: **for** iter **in** iterations **do**
3:    Sample a mini-batch of samples $x_1, ..., x_B$ forming the empirical batch distribution $\mathbb{P}_b$
4:    Encode: $z_{i \to B} = f_e(x_{i \to B})$     // $i \to B$ : for $i = 1, ..., B$
5:    Quantize: $c_{i \to B} = \arg\min_k d_z(z_{i \to B}, c_k)$     // *Nearest neighbor assignment*
6:    Decode: $\tilde{x}_{i \to B} = f_d(c_{i \to B})$
7:    Optimize $f_e, f_d$ and $C$ by minimizing the objective in (4):

$$\frac{1}{B} \sum_{i=1}^{B} [d_x(\tilde{x}_i, x_i)] + \lambda \underbrace{\mathcal{W}_{d_z}(f_e \# \mathbb{P}_b, \mathbb{P}_{c,\pi})}_{\min_{P \in \Gamma(1_B, 1_C)} \langle P, D_{c,x} \rangle}$$

8: **end for**
9: **Return:** The optimal $f_e, f_d$ and $C$.

---

## 4 RELATED WORK

Variational Auto-Encoder (VAE) was first introduced by Kingma & Welling (Kingma & Welling, 2013) for learning continuous representations. However, learning discrete latent representations has

proved much more challenging because it is nearly impossible to accurately evaluate the gradients which are required to train models. To make the gradients tractable, one possible solution is to apply the Gumbel Softmax reparameterization trick (Jang et al., 2016) to VAE, which allows us to estimate stochastic gradients for updating the models. Although this technique has a low variance, it brings up a high-bias gradient estimator. Another possible solution is to employ the REINFORCE algorithm (Williams, 1992), which is unbiased but has a high variance. Additionally, the two techniques can be complementarily combined (Tucker et al., 2017).

To enable learning the discrete latent codes, VQ-VAE (Van Den Oord et al., 2017) uses deterministic encoder/decoder and encourages the codebooks to become the clustering centroids of latent representations. Additionally, the copy gradient trick is employed in back-propagating gradients from the decoder to the encoder (Bengio, 2013). Some further works were proposed to extend VQ-VAE, notably (Roy et al., 2018; Wu & Flierl, 2020). Particularly, Roy et al. (2018) uses the Expectation Maximization (EM) algorithm in the bottleneck stage to train the VQ-VAE for improving the quality of the generated images. However, to maintain the stability of this approach, we need to collect a large number of samples on the latent space. Wu & Flierl (2020) imposes noises on the latent codes and uses a Bayesian estimator to optimize the quantizer-based representation. The introduced bottleneck Bayesian estimator outputs the posterior mean of the centroids to the decoder and performs soft quantization of the noisy latent codes which have latent representations preserving the similarity relations of the data space. Recently, Takida et al. (2022) extends the standard VAE with stochastic quantization and trainable posterior categorical distribution, showing that the annealing of the stochasticity of the quantization process significantly improves the codebook utilization.

Wasserstein (WS) distance has been widely used in generative models (Arjovsky et al., 2017; Gulrajani et al., 2017; Tolstikhin et al., 2017). Arjovsky et al. Arjovsky et al. (2017) uses a dual form of WS distance to develop Wasserstein generative adversarial network (WGAN). Later, Gulrajani et al. (2017) employs the gradient penalty trick to improve the stability of WGAN. In terms of theory development, mostly related to our work is Wasserstein Auto-Encoder (Tolstikhin et al., 2017) which aims to learn continuous latent representation preserving the characteristics of input data.

## 5 EXPERIMENTS

In this section, we conduct extensive experiments to show the effectiveness of our proposed method compared to other advances.

**Datasets**: we empirically evaluate the proposed VQ-WAE in comparison with VQ-VAE (Van Den Oord et al., 2017) that is the baseline method and recently proposed SQ-VAE (Takida et al., 2022) which is the state-of-the-art work of improving the codebook usage, on four different benchmark datasets: CIFAR10 (Van Den Oord et al., 2017), MNIST (Deng, 2012), SVHN (Netzer et al., 2011), CelebA (Liu et al., 2015) and the high-resolution images dataset FFHQ Karras et al. (2019).

**Implementation**: For a fair comparison, we utilize the same architectures and hyper-parameters for all methods. Additionally, in the primary setting, we use the codeword (discrete latent) dimensionality of 64 and codebook size $|C| = 512$ for all datasets except FFHQ with codeword dimensionality of 256 and $|C| = 1024$, while the hyper-parameters $\{\beta, \tau, \lambda\}$ are specified as presented in the original papers, *i.e.*, $\beta = 0.25$ for VQ-VAE and VQ-GAN (Esser et al., 2021), $\tau = 1e^{-5}$ for SQ-VAE and $\lambda = 1$ for our VQ-WAE. The details of the experimental settings are presented in Appendix C.

### 5.1 RESULTS ON BENCHMARK DATASETS

In order to quantitatively assess the quality of the reconstructed images, we report the results on most common evaluation metrics, including the pixel-level peak signal-to-noise ratio (PSNR), patch-level structure similarity index (SSIM), feature-level LPIPS (Zhang et al., 2018), and dataset-level Fréchlet Inception Distance (FID) (Heusel et al., 2017). We report the test-set reconstruction results on four datasets in Table 1. With regard to the codebook utilization, we employ perplexity score which is defined as $e^{-\sum_{k=1}^{K} p_{c_k} \log p_{c_k}}$ where $p_{c_k} = \frac{N_{c_k}}{\sum_{i=1}^{K} N_{c_i}}$ (i.e., $N_{c_i}$ is the number of latent representations associated with the codeword $c_i$) is the probability of the $i^{th}$ codeword being used. Note that by formula, perplexity$_{\max} = |C|$ as $P(c)$ becomes to the uniform distribution, which means that all the codewords are utilized equally by the model.

Table 1: Reconstruction performance (↓: the lower the better and ↑: the higher the better).

| Dataset | Model | Latent Size | SSIM ↑ | PSNR ↑ | LPIPS ↓ | rFID ↓ | Perplexity ↑ |
|---------|-------|-------------|--------|--------|---------|--------|--------------|
| CIFAR10 | VQ-VAE | $8 \times 8$ | 0.70 | 23.14 | 0.35 | 77.3 | 69.8 |
|  | SQ-VAE | $8 \times 8$ | **0.80** | **26.11** | **0.23** | 55.4 | 434.8 |
|  | VQ-WAE | $8 \times 8$ | **0.80** | 25.93 | **0.23** | **54.9** | **505.0** |
| MNIST | VQ-VAE | $8 \times 8$ | 0.98 | 33.37 | 0.02 | 4.8 | 47.2 |
|  | SQ-VAE | $8 \times 8$ | **0.99** | **36.25** | **0.01** | 3.2 | 301.8 |
|  | VQ-WAE | $8 \times 8$ | **0.99** | 35.61 | **0.01** | **2.4** | **507.7** |
| SVHN | VQ-VAE | $8 \times 8$ | 0.88 | 26.94 | 0.17 | 38.5 | 114.6 |
|  | SQ-VAE | $8 \times 8$ | **0.96** | **35.37** | **0.06** | 24.8 | 389.8 |
|  | VQ-WAE | $8 \times 8$ | **0.96** | 34.67 | **0.06** | **22.6** | **486.0** |
| CELEBA | VQ-VAE | $16 \times 16$ | 0.82 | 27.48 | 0.19 | 19.4 | 48.9 |
|  | SQ-VAE | $16 \times 16$ | **0.89** | **31.05** | **0.12** | 14.8 | 427.8 |
|  | VQ-WAE | $16 \times 16$ | 0.88 | 30.08 | 0.13 | **13.6** | **508.0** |
| FFHQ | VQ-GAN | $16 \times 16$ | 0.6641 | 22.24 | **0.12** | 4.42 | 423 |
|  | VQ-WAE | $16 \times 16$ | **0.6648** | **22.45** | 0.1245 | **4.20** | **1022** |

We compare VQ-WAE with VQ-VAE, SQ-VAE and VQ-GAN for image reconstruction in Table 1. All instantiations of our model significantly outperform the baseline VQ-VAE under the same compression ratio, with the same network architecture. While the latest state-of-the-art SQ-VAE holds slightly better scores for traditional pixel- and patch-level metrics, our method achieves much better rFID scores which evaluate the image quality at the dataset level. Note that our VQ-WAE significantly improves the perplexity of the learned codebook. This suggests that the proposed method significantly improves the codebook usage, resulting in better reconstruction quality. Finally, to complete the assessment, the qualitative results are visualized in Figure 4 (Appendix B).

## 5.2 DETAILED ANALYSIS

We run a number of ablations to analyze the properties of VQ-VAE, SQ-VAE and VQ-WAE, in order to assess if our VQ-WAE can simultaneously achieve (i) efficient codebook usage, (ii) reasonable latent representation.

### 5.2.1 CODEBOOK USAGE

Table 2: Distortion and Perplexity with different codebook sizes.

| Dataset |  | MNIST |  |  |  | CIFAR10 |  |  |  |
|---------|--|-------|---|---|---|---------|---|---|---|
| $\|C\|$ |  | 64 | 128 | 256 | 512 | 64 | 128 | 256 | 512 |
| VQ-VAE | Perplexity | 47.8 | 70.3 | 52.0 | 47.2 | 24.3 | 44.9 | 85.1 | 69.8 |
|  | rFID | 5.9 | 6.2 | 5.2 | 4.8 | 86.6 | 78.9 | 73.6 | 69.8 |
| SQ-VAE | Perplexity | 47.4 | 85.4 | 184.8 | 301.8 | 59.5 | 113.2 | 220.0 | 434.8 |
|  | rFID | **4.7** | 4.3 | 3.5 | 3.2 | **71.5** | **66.9** | 62.6 | 55.4 |
| VQ-WAE | Perplexity | **63.8** | **127.7** | **255.1** | **507.7** | **63.4** | **126.1** | **252.0** | **505.0** |
|  | rFID | 5.6 | **3.8** | **2.8** | **2.4** | 73.5 | 68.5 | **60.3** | **54.9** |

We observe the codebook utilization of three methods with different codebook sizes $\{64, 128, 256, 512\}$ on MNIST and CIFAR10 datasets. Particularly, we present the reconstruction performance for different settings in Table 2 and the histogram of latent representations over the codebook in Figure 2.

As discussed in Section 3.1 and Section 3.2, the number of used centroids reflects the capability of the latent representations. In other words, it represents the certain amount of information is pre-

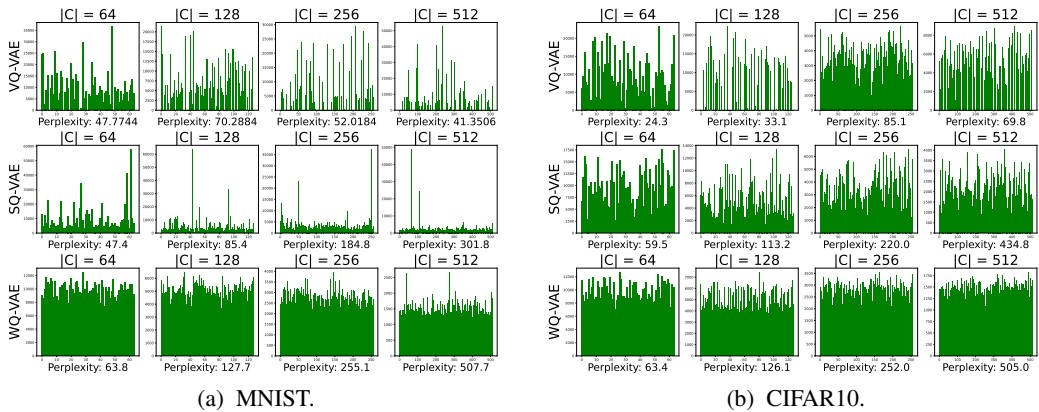

Figure 2: Latent distribution over the codebook on test-set.

served in the latent space. By explicitly defining the numbers of latent representations associated with the codebooks to be uniform (i.e., fixing $\pi$ in (4) as a uniform distribution) in the Wasserstein regularization term, VQ-WAE is able to maximize the information in the codebooks, hence improving the reconstruction capacity. It can be seen from Figure 2 that the latent distribution of VQ-WAE over the codebook is nearly uniform and the codebook's perplexity almost reaches the optimal value (i.e., the value of perplexities reach to corresponding codebook sizes) in different settings. *It is also observed that as the size of the codebook increases, the perplexity of codebook of VQ-WAE also increases, leading to the better reconstruction performance (Table 2), in line with the analysis in (Wu & Flierl, 2018).* SQ-VAE also has good codebook utilization as its perplexity is proportional to the size of the codebook. However, its codebook utilization becomes less efficient when the codebook size becomes large, especially in low texture dataset (i.e., MNIST).

On the contrary, the codebook usage of VQ-VAE is less efficient, i.e., there are many zero entries in its codebook usage histogram, indicating that some codewords have never been used (Figure 2). Furthermore, Table 2 also shows the instability of VQ-VAE's reconstruction performance with different codebook sizes.

### 5.2.2 VISUALIZATION OF LATENT REPRESENTATION

To better understand the codebook's representation power, we employ t-SNE (van der Maaten & Hinton, 2008) to visualize the latent representations that have been learned by VQ-VAE, SQ-VAE and VQ-WAE on the MNIST dataset with two codebook sizes of 64 and 512. Figure 3 shows the latent distributions of different classes in the latent space, in which the samples are colored accordingly to their class labels. Figure 3c shows that representations from different classes of VQ-WAE are well clustered (i.e., each class focuses on only one cluster) and clearly separated to other classes. In contrast, the representations of some classes in VQ-VAE and SQ-VAE are distributed to several clusters and or mixed to each other (Figure 3a,b). Moreover, the class-clusters of SQ-VAE are uncondensed and tend to overlap with each other. These results suggest that the representations learned by VQ-WAE can better preserve the similarity relations of the data space better than the other models.

### 5.2.3 IMAGE GENERATION

As discussed in the previous section, VQ-WAE is able to optimally utilize its codebook, leading to meaningful and diverse codewords that naturally improve the image generation. To confirm this ability, we perform the image generation on the benchmark datasets. Since the decoder reconstructs images directly from the discrete embeddings, we only need to model a prior distribution over the discrete latent space (i.e., codebook) to generate images.

We employ a conventional autoregressive model, the CNN-based PixelCNN (Van den Oord et al., 2016), to estimate a prior distribution over the discrete latent space of VQ-VAE, SQ-VAE and VQ-WAE on CIFAR10, MNIST, SVHN and CelebA. The details of generation settings are presented

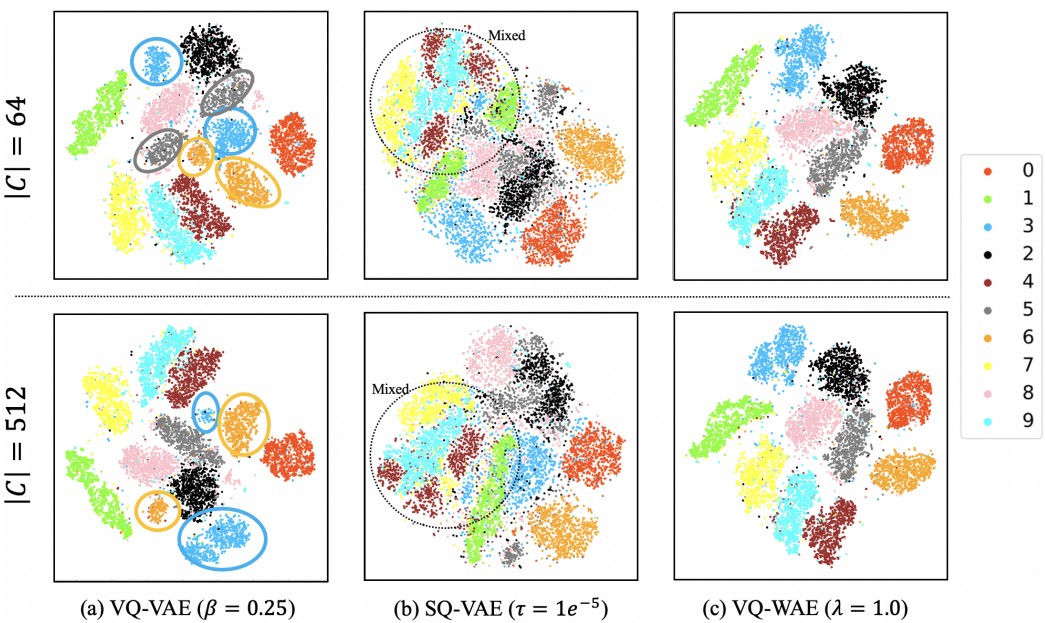

Figure 3: The t-SNE feature visualization on the MNIST dataset.

Table 3: FID scores of generated images.

| Dataset | VQ-Model | Generation | Latent Size | $|C|$ | unconditional | class-conditional |
|---------|----------|------------|-------------|-------|---------------|-------------------|
| CIFAR10 | VQ-VAE | PixelCNN | $8 \times 8$ | 512 | 117.49 | 117.16 |
| | SQ-VAE | PixelCNN | $8 \times 8$ | 512 | 103.78 | 90.74 |
| | VQ-WAE | PixelCNN | $8 \times 8$ | 512 | **87.62** | **88.93** |
| MNIST | VQ-VAE | PixelCNN | $8 \times 8$ | 512 | 27.01 | 25.56 |
| | SQ-VAE | PixelCNN | $8 \times 8$ | 512 | 8.93 | 4.94 |
| | VQ-WAE | PixelCNN | $8 \times 8$ | 512 | **8.17** | **3.96** |
| SVHN | VQ-VAE | PixelCNN | $8 \times 8$ | 512 | 62.13 | 64.24 |
| | SQ-VAE | PixelCNN | $8 \times 8$ | 512 | 31.26 | 36.41 |
| | VQ-WAE | PixelCNN | $8 \times 8$ | 512 | **30.64** | **34.24** |
| CELEBA | VQ-VAE | PixelCNN | $16 \times 16$ | 512 | 42.0 | - |
| | SQ-VAE | PixelCNN | $16 \times 16$ | 512 | 29.5 | - |
| | VQ-WAE | PixelCNN | $16 \times 16$ | 512 | **28.8** | - |

in Section 3.2 of the supplementary material. The quantitative results in Table 3 indicate that the codebook of VQ-WAE leads to a better generation ability than VQ-VAE and SQ-VAE.

## 6    CONCLUSION

In this paper, inspired by the nice properties and mature theory of the WS distance allowing it to be applied successfully to generative models and continous representation learning, we propose Vector Quantized Wasserstein Auto-Encoder (VQ-WAE), which endows a discrete distribution over the codewords and learns a deterministic decoder that transports the codeword distribution to the data distribution via minimizing a WS distance between them. We then developed theoretical analysis to show the equivalence of this WS minimization to another OP regarding push-forwarding the data distribution to the codeword distribution, which can be realized by minimizing a WS distance between the latent representation and codeword distributions. We conduct comprehensive experiments to show that our VQ-WAE utilizes the codebooks more efficiently than the baselines, hence leading to better reconstructed and generated image quality.

## 7 REPRODUCIBILITY STATEMENT

We provide the implementation of our framework in the supplementary material.

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

APPENDIX

This appendix is organized as follows:

- In Section A, we present all proofs for theory developed in the main paper.
- In Section B, we present additional experimental results on the high-quality image dataset FFHQ.
- In Section C, we present experimental settings and implementation specification of VQ-WAE.

## A  THEORETICAL DEVELOPMENT

Given a training set $\mathbb{D} = \{x_1, ..., x_N\} \subset \mathbb{R}^V$, we wish to learn a *set of codebooks* $C = \{c_k\}_{k=1}^K \in \mathbb{R}^{K \times D}$ on a latent space $\mathcal{Z}$ and *an encoder* to map each data example to a given codebook, preserving insightful characteristics carried in data. We first endow a discrete distribution over the codebooks as $\mathbb{P}_{c,\pi} = \sum_{k=1}^K \pi_k \delta_{c_k}$ with the Dirac delta function $\delta$ and the weights $\pi \in \Delta_K$. We aim to learn a decoder function $f_d : \mathcal{Z} \to \mathcal{X}$ (i.e., mapping from the latent space $\mathcal{Z} \subset \mathbb{R}^D$ to the data space $\mathcal{X}$), the codebooks $C$, and the weights $\pi$, to minimize:

$$\min_{C, \pi} \min_{f^d} \mathcal{W}_{d_x} \left( f_d \# \mathbb{P}_{c,\pi}, \mathbb{P}_x \right), \tag{7}$$

where $\mathbb{P}_x = \frac{1}{N} \sum_{n=1}^N \delta_{x_n}$ is the empirical data distribution and $d_x$ is a cost metric on the data space.

**Lemma A.1.** *(Lemma 3.1 in the main paper) Let $C^* = \{c_k^*\}_k$, $\pi^*$, and $f_d^*$ be the optimal solution of the OP in Eq. (7). Assume $K < N$, then $C^* = \{c_k^*\}_k$, $\pi^*$, and $f_d^*$ are also the optimal solution of the following OP:*

$$\min_{f_d} \min_{C, \pi} \min_{\sigma \in \Sigma_\pi} \sum_{n=1}^N d_x \left( x_n, f_d \left( c_{\sigma(n)} \right) \right), \tag{8}$$

*where $\Sigma_\pi$ is the set of assignment functions $\sigma : \{1, ..., N\} \to \{1, ..., K\}$ such that the cardinalities $\left| \sigma^{-1}(k) \right|, k = 1, ..., K$ are proportional to $\pi_k, k = 1, ..., K$.*

**Proof of Lemma** A.1

It is clear that

$$f_d \# \mathbb{P}_{c,\pi} = \sum_{k=1}^K \pi_k \delta_{f_d(c_k)}.$$

Therefore, we reach the following OP:

$$\min_{C, \pi} \min_{f^d} \mathcal{W}_{d_x} \left( \frac{1}{N} \sum_{n=1}^N \delta_{x_n}, \sum_{k=1}^K \pi_k \delta_{f_d(c_k)} \right). \tag{9}$$

By using the Monge definition, we have

$$\mathcal{W}_{d_x} \left( \frac{1}{N} \sum_{n=1}^N \delta_{x_n}, \sum_{k=1}^K \pi_k \delta_{f_d(c_k)} \right) = \min_{T : T \# \mathbb{P}_x = f_d \# \mathbb{P}_{c,\pi}} \mathbb{E}_{x \sim \mathbb{P}_x} \left[ d_x \left( x, T(x) \right) \right]$$

$$= \frac{1}{N} \min_{T : T \# \mathbb{P}_x = f_d \# \mathbb{P}_{c,\pi}} \sum_{n=1}^N d_x \left( x_n, T(x_n) \right).$$

Since $T \# \mathbb{P}_x = f_d \# \mathbb{P}_{c,\pi}$, $T(x_n) = f_d(c_k)$ for some $k$. Additionally, $\left| T^{-1}(f_d(c_k)) \right|, k = 1, ..., K$ are proportional to $\pi_k, k = 1, ..., K$. Denote $\sigma : \{1, ..., N\} \to \{1, ..., K\}$ such that $T(x_n) = f_d(c_{\sigma(n)}), \forall i = 1, ..., N$, we have $\sigma \in \Sigma_\pi$. It follows that

$$\mathcal{W}_{d_x} \left( \frac{1}{N} \sum_{n=1}^N \delta_{x_n}, \sum_{k=1}^K \pi_k \delta_{f_d(c_k)} \right) = \frac{1}{N} \min_{\sigma \in \Sigma_\pi} \sum_{n=1}^N d_x \left( x_n, f_d \left( c_{\sigma(n)} \right) \right).$$

Finally, the the optimal solution of the OP in Eq. (7) is equivalent to

$$\min_{f_d} \min_{C,\pi} \min_{\sigma \in \Sigma_\pi} \sum_{n=1}^{N} d_x \left( x_n, f_d \left( c_{\sigma(n)} \right) \right),$$

which directly implies the conclusion.

**Theorem A.1.** *(Theorem 3.1 in the main paper) We can equivalently turn the optimization problem in (7) to*

$$\min_{C,\pi,f_d} \min_{\bar{f}_e : \bar{f}_e \# \mathbb{P}_x = \mathbb{P}_{c,\pi}} \mathbb{E}_{x \sim \mathbb{P}_x} \left[ d_x \left( f_d \left( \bar{f}_e \left( x \right) \right), x \right) \right], \tag{10}$$

*where $\bar{f}_e$ is a **deterministic discrete** encoder mapping data example $x$ directly to the codebooks.*

**Proof of Theorem A.1**

We first prove that the OP of interest in (7) is equivalent to

$$\min_{C,\pi,f_d} \min_{\bar{f}_e : \bar{f}_e \# \mathbb{P}_x = \mathbb{P}_{c,\pi}} \mathbb{E}_{x \sim \mathbb{P}_x, c \sim \bar{f}_e(x)} \left[ d_x \left( f_d \left( c \right), x \right) \right], \tag{11}$$

where $\bar{f}_e$ is a **stochastic discrete** encoder mapping data example $x$ directly to the codebooks. To this end, we prove that

$$\mathcal{W}_{d_x} \left( f_d \# \mathbb{P}_{c,\pi}, \mathbb{P}_x \right) = \min_{\bar{f}_e : \bar{f}_e \# \mathbb{P}_x = \mathbb{P}_{c,\pi}} \mathbb{E}_{x \sim \mathbb{P}_x, c \sim \bar{f}_e(x)} \left[ d_x \left( f_d \left( c \right), x \right) \right], \tag{12}$$

where $\bar{f}_e$ is a **stochastic discrete** encoder mapping data example $x$ directly to the codebooks.

Let $\bar{f}_e$ be a **stochastic discrete** encoder such that $\bar{f}_e \# \mathbb{P}_x = \mathbb{P}_{c,\pi}$ (i.e., $x \sim \mathbb{P}_x$ and $c \sim \bar{f}_e(x)$ implies $c \sim \mathbb{P}_{c,\pi}$). We consider $\gamma_{d,c}$ as the joint distribution of $(x,c)$ with $x \sim \mathbb{P}_x$ and $c \sim \bar{f}_e(x)$. We also consider $\gamma_{fc,d}$ as the joint distribution including $(x,x') \sim \gamma_{fc,d}$ where $x \sim \mathbb{P}_x, c \sim \bar{f}_e(x)$, and $x' = f_d(c)$. This follows that $\gamma_{fc,d} \in \Gamma \left( f_d \# \mathbb{P}_{c,\pi}, \mathbb{P}_x \right)$ which admits $f_d \# \mathbb{P}_{c,\pi}$ and $\mathbb{P}_x$ as its marginal distributions. We also have:

$$\mathbb{E}_{x \sim \mathbb{P}_x, c \sim \bar{f}_e(x)} \left[ d_x \left( f_d \left( c \right), x \right) \right] = \mathbb{E}_{(x,c) \sim \gamma_{d,c}} \left[ d_x \left( f_d \left( c \right), x \right) \right] \overset{(1)}{=} \mathbb{E}_{(x,x') \sim \gamma_{fc,d}} \left[ d_x \left( x, x' \right) \right]$$

$$\geq \min_{\gamma_{fc,d} \in \Gamma(f_d \# \mathbb{P}_{c,\pi}, \mathbb{P}_x)} \mathbb{E}_{(x,x') \sim \gamma_{fc,d}} \left[ d_x \left( x, x' \right) \right]$$

$$= \mathcal{W}_{d_x} \left( f_d \# \mathbb{P}_{c,\pi}, \mathbb{P}_x \right).$$

Note that we have the equality in (1) due to $(id, f_d) \# \gamma_{d,c} = \gamma_{fc,d}$.

Therefore, we reach

$$\min_{\bar{f}_e : \bar{f}_e \# \mathbb{P}_x = \mathbb{P}_{c,\pi}} \mathbb{E}_{x \sim \mathbb{P}_x, c \sim \bar{f}_e(x)} \left[ d_x \left( f_d \left( c \right), x \right) \right] \geq \mathcal{W}_{d_x} \left( f_d \# \mathbb{P}_{c,\pi}, \mathbb{P}_x \right).$$

Let $\gamma_{fc,d} \in \Gamma \left( f_d \# \mathbb{P}_{c,\pi}, \mathbb{P}_x \right)$. Let $\gamma_{fc,c} \in \Gamma \left( f_d \# \mathbb{P}_{c,\pi}, \mathbb{P}_{c,\pi} \right)$ be a deterministic coupling such that $c \sim \mathbb{P}_{c,\pi}$ and $x = f_d(c)$ imply $(c,x) \sim \gamma_{c,fc}$. Using the gluing lemma (see Lemma 5.5 in Santambrogio (2015)), there exists a joint distribution $\alpha \in \Gamma \left( \mathbb{P}_{c,\pi}, f_d \# \mathbb{P}_{c,\pi}, \mathbb{P}_x \right)$ which admits $\gamma_{fc,d}$ and $\gamma_{fc,c}$ as the corresponding joint distributions. By denoting $\gamma_{d,c} \in \Gamma \left( \mathbb{P}_x, \mathbb{P}_{c,\pi} \right)$ as the marginal distribution of $\alpha$ over $\mathbb{P}_x, \mathbb{P}_{c,\pi}$, we then have

$$\mathbb{E}_{(x,x') \sim \gamma_{fc,d}} \left[ d_x \left( x, x' \right) \right] = \mathbb{E}_{(c,x',x) \sim \alpha} \left[ d_x \left( x, x' \right) \right] = \mathbb{E}_{(c,x) \sim \gamma_{d,c}, x' = f_d(c)} \left[ d_x \left( x, x' \right) \right]$$

$$= \mathbb{E}_{(c,x) \sim \gamma_{d,c}} \left[ d_x \left( f_d \left( c \right), x \right) \right] = \mathbb{E}_{x \sim \mathbb{P}_x, c \sim \bar{f}_e(x)} \left[ d_x \left( f_d \left( c \right), x \right) \right].$$

$$\geq \min_{\bar{f}_e : \bar{f}_e \# \mathbb{P}_x = \mathbb{P}_{c,\pi}} \mathbb{E}_{x \sim \mathbb{P}_x, c \sim \bar{f}_e(x)} \left[ d_x \left( f_d \left( c \right), x \right) \right],$$

where $\bar{f}_e(x) = \gamma_{d,c}(\cdot \mid x)$.

This follows that

$$\mathcal{W}_{d_x} \left( f_d \# \mathbb{P}_{c,\pi}, \mathbb{P}_x \right) = \min_{\gamma_{fc,d} \in \Gamma(f_d \# \mathbb{P}_{c,\pi}, \mathbb{P}_x)} \mathbb{E}_{(x,x') \sim \gamma_{fc,d}} \left[ d_x \left( x, x' \right) \right]$$

$$\geq \min_{\bar{f}_e : \bar{f}_e \# \mathbb{P}_x = \mathbb{P}_{c,\pi}} \mathbb{E}_{x \sim \mathbb{P}_x, c \sim \bar{f}_e(x)} \left[ d_x \left( f_d \left( c \right), x \right) \right].$$

This completes the proof for the equality in Eq. (12), which means that the OP of interest in (7) is equivalent to

$$\min_{C,\pi,f_d} \min_{\bar{f}_e : \bar{f}_e \# \mathbb{P}_x = \mathbb{P}_{c,\pi}} \mathbb{E}_{x \sim \mathbb{P}_x, c \sim \bar{f}_e(x)} \left[ d_x \left( f_d \left( c \right), x \right) \right], \tag{13}$$

We now further prove the above OP is equivalent to

$$\min_{C,\pi,f_d} \min_{\bar{f}_e : \bar{f}_e \# \mathbb{P}_x = \mathbb{P}_{c,\pi}} \mathbb{E}_{x \sim \mathbb{P}_x} \left[ d_x \left( f_d \left( \bar{f}_e \left( x \right) \right), x \right) \right], \tag{14}$$

where $\bar{f}_e$ is a **deterministic discrete** encoder mapping data example $x$ directly to the codebooks.

It is obvious that the OP in (14) is special case of that in (13) when we limit to search for deterministic discrete encoders. Given the optimal solution $C^{*1}, \pi^{*1}, f_d^{*1}$, and $\bar{f}_e^{*1}$ of the OP in (13), we show how to construct the optimal solution for the OP in (14). Let us construct $C^{*2} = C^{*1}$, $f_d^{*2} = f_d^{*1}$. Given $x \sim \mathbb{P}_x$, let us denote $\bar{f}_e^{*2}(x) = \mathrm{argmin}_c d_x \left( f_d^{*2} \left( c \right), x \right)$. Thus, $\bar{f}_e^{*2}$ is a deterministic discrete encoder mapping data example $x$ directly to the codebooks. We define $\pi_k^{*2} = Pr \left( \bar{f}_e^{*2} \left( x \right) = c_k : x \sim \mathbb{P}_x \right), k = 1, ..., K$, meaning that $\bar{f}_e^{*2} \# \mathbb{P}_x = \mathbb{P}_{c^{*2}, \pi^{*2}}$. From the construction of $\bar{f}_e^{*2}$, we have

$$\mathbb{E}_{x \sim \mathbb{P}_x} \left[ d_x \left( f_d^{*2} \left( \bar{f}_e^{*2} \left( x \right) \right), x \right) \right] \leq \mathbb{E}_{x \sim \mathbb{P}_x, c \sim \bar{f}_e^{*1}(x)} \left[ d_x \left( f_d^{*1} \left( c \right), x \right) \right].$$

Furthermore, because $C^{*2}, \pi^{*2}, f_d^{*2}$, and $\bar{f}_e^{*2}$ are also a feasible solution of the OP in (14), we have

$$\mathbb{E}_{x \sim \mathbb{P}_x} \left[ d_x \left( f_d^{*2} \left( \bar{f}_e^{*2} \left( x \right) \right), x \right) \right] \geq \mathbb{E}_{x \sim \mathbb{P}_x, c \sim \bar{f}_e^{*1}(x)} \left[ d_x \left( f_d^{*1} \left( c \right), x \right) \right].$$

This means that

$$\mathbb{E}_{x \sim \mathbb{P}_x} \left[ d_x \left( f_d^{*2} \left( \bar{f}_e^{*2} \left( x \right) \right), x \right) \right] = \mathbb{E}_{x \sim \mathbb{P}_x, c \sim \bar{f}_e^{*1}(x)} \left[ d_x \left( f_d^{*1} \left( c \right), x \right) \right],$$

and $C^{*2}, \pi^{*2}, f_d^{*2}$, and $\bar{f}_e^{*2}$ are also the optimal solution of the OP in (14).

Additionally, $\bar{f}_e$ is a deterministic discrete encoder mapping data example $x$ directly to the codebooks. To make it trainable, we replace $\bar{f}_e$ by a continuous encoder $f_e : \mathcal{X} \to \mathcal{Z}$ and arrive the following OP:

$$\min_{C,\pi} \min_{f_d, f_e} \left\{ \mathbb{E}_{x \sim \mathbb{P}_x} \left[ d_x \left( f_d \left( Q_C \left( f_e \left( x \right) \right) \right), x \right) \right] + \lambda \mathcal{W}_{d_z} \left( f_e \# \mathbb{P}_x, \mathbb{P}_{c,\pi} \right) \right\}, \tag{15}$$

where $Q_C \left( f_e \left( x \right) \right) = \mathrm{argmin}_{c \in C} d_z \left( f_e \left( x \right), c \right)$ is a *quantization operator* which returns the closest codebook to $f_e \left( x \right)$ and the parameter $\lambda > 0$.

We now propose and prove the following lemma that is necessary for the proof of Theorem A.2.

**Lemma A.2.** *Consider* $C, \pi, f_d$, *and* $f_e$ *as a feasible solution of the OP in (15). Let us denote* $\bar{f}_e(x) = argmin_c d_z(f_e(x)), c) = Q_C(x)$, *then* $\bar{f}_e(x)$ *is a Borel measurable function.*

**Proof of Lemma A.2**.

We denote the set $A_k$ on the latent space as

$$A_k = \{z : d_z(z, c_k) < d(z, c_j), \forall j \neq k\} = \{z : Q_C(z) = c_k\}.$$

$A_k$ is known as a Voronoi cell w.r.t. the metric $d_z$. If we consider a continuous metric $d_z$, $A_k$ is a measurable set. Given a Borel measurable function $B$, we prove that $\bar{f}_e^{-1}(B)$ is a Borel measurable set on the data space.

Let $B \cap \{c_1, .., c_K\} = \{c_{i_1}, ..., c_{i_m}\}$, we prove that $\bar{f}_e^{-1}(B) = \cup_{j=1}^m f_e^{-1}(A_{i_j})$. Indeed, take $x \in \bar{f}_e^{-1}(B)$, then $\bar{f}_e(x) \in B$, implying that $\bar{f}_e(x) = Q_C(x) = c_{i_j}$ for some $j = 1, ..., m$. This means that $f_e(x) \in A_{i_j}$ for some $j = 1, ..., m$. Therefore, we reach $\bar{f}_e^{-1}(B) \subset \cup_{j=1}^m f_e^{-1}(A_{i_j})$.

We now take $x \in \cup_{j=1}^m f_e^{-1}(A_{i_j})$. Then $f_e(x) \in A_{i_j}$ for $j = 1, ..., m$, hence $\bar{f}_e(x) = Q_C(x) = c_{i_j}$ for some $j = 1, ..., m$. Thus, $\bar{f}_e(x) \subset B$ or equivalently $x \in \bar{f}_e^{-1}(B)$, implying $\bar{f}_e^{-1}(B) \supset \cup_{j=1}^m f_e^{-1}(A_{i_j})$.

Finally, we reach $\bar{f}_e^{-1}(B) = \cup_{j=1}^m f_e^{-1}(A_{i_j})$, which concludes our proof because $f_e$ is a measurable function and $A_{i_j}$ are measurable sets.

**Theorem A.2.** *(Theorem 3.2 in the main paper) If we seek $f_d$ and $f_e$ in a family with infinite capacity (e.g., the space of all measurable functions), the three OPs of interest in (7, 10, and 15) are equivalent.*

**Proof of Theorem A.2**.

Given the optimal solution $C^{*1}, \pi^{*1}, f_d^{*1}$, and $f_e^{*1}$ of the OP in (15), we conduct the optimal solution for the OP in (10). Let us conduct $C^{*2} = C^{*1}, f_d^{*2} = f_d^{*1}$. We next define $\bar{f}_e^{*2}(x) = \mathrm{argmin}_c d_z \left( f_e^{*1}(x), c \right) = Q_{C^{*1}} \left( f_e^{*1}(x) \right) = Q_{C^{*2}} \left( f_e^{*1}(x) \right)$. We prove that $C^{*2}, \pi^{*2}, f_d^{*2}$, and $\bar{f}_e^{*2}$ are optimal solution of the OP in (10). By this definition, we yield $\bar{f}_e^{*2} \# \mathbb{P}_x = \mathbb{P}_{c^{*2}, \pi^{*2}}$ and hence $\mathcal{W}_{d_z} \left( \bar{f}_e^{*2} \# \mathbb{P}_x, \mathbb{P}_{c^{*2}, \pi^{*2}} \right) = 0$. Therefore, we need to verify the following:

(i) $\bar{f}_e^{*2}$ is a Borel-measurable function.

(ii) Given a feasible solution $C, \pi, f_d$, and $\bar{f}_e$ of (10), we have

$$\mathbb{E}_{x \sim \mathbb{P}_x} \left[ d_x \left( f_d^{*2} \left( \bar{f}_e^{*2}(x) \right), x \right) \right] \leq \mathbb{E}_{x \sim \mathbb{P}_x} \left[ d_x \left( f_d \left( \bar{f}_e(x) \right), x \right) \right]. \tag{16}$$

We first prove (i). It is a direct conclusion because the application of Lemma A.2 to $C^{*1}, \pi^{*1}, f_d^{*1}$, and $f_e^{*1}$.

We next prove (ii). We further derive as

$$\begin{aligned}
&\mathbb{E}_{x \sim \mathbb{P}_x} \left[ d_x \left( f_d^{*2} \left( \bar{f}_e^{*2}(x) \right), x \right) \right] + \lambda \mathcal{W}_{d_z} \left( \bar{f}_e^{*2} \# \mathbb{P}_x, \mathbb{P}_{c^{*2}, \pi^{*2}} \right) \\
&= \mathbb{E}_{x \sim \mathbb{P}_x} \left[ d_x \left( f_d^{*2} \left( \bar{f}_e^{*2}(x) \right), x \right) \right] \\
&= \mathbb{E}_{x \sim \mathbb{P}_x} \left[ d_x \left( f_d^{*1} \left( Q_{C^{*2}} \left( f_e^{*1}(x) \right) \right), x \right) \right] \\
&= \mathbb{E}_{x \sim \mathbb{P}_x} \left[ d_x \left( f_d^{*1} \left( Q_{C^{*1}} \left( f_e^{*1}(x) \right) \right), x \right) \right] \\
&\leq \mathbb{E}_{x \sim \mathbb{P}_x} \left[ d_x \left( f_d^{*1} \left( Q_{C^{*1}} \left( f_e^{*1}(x) \right) \right), x \right) \right] + \lambda \mathcal{W}_{d_z} \left( f_e^{*1} \# \mathbb{P}_x, \mathbb{P}_{c^{*1}, \pi^{*1}} \right). \tag{17}
\end{aligned}$$

Moreover, because $\bar{f}_e \# \mathbb{P}_x = \mathbb{P}_{c, \pi}$ which is a discrete distribution over the set of codewords $C$, we obtain $Q_C(\bar{f}_e(x)) = \bar{f}_e(x)$. Note that $C, \pi, f_d$, and $\bar{f}_e$ is also a feasible solution of (15) because $\bar{f}_e$ is also a specific encoder mapping from the data space to the latent space, we achieve

$$\begin{aligned}
&\mathbb{E}_{x \sim \mathbb{P}_x} \left[ d_x \left( f_d \left( Q_C \left( \bar{f}_e(x) \right) \right), x \right) \right] + \lambda \mathcal{W}_{d_z} \left( \bar{f}_e \# \mathbb{P}_x, \mathbb{P}_{c, \pi} \right) \\
&\geq \mathbb{E}_{x \sim \mathbb{P}_x} \left[ d_x \left( f_d^{*1} \left( Q_{C^{*1}} \left( \bar{f}_e^{*1}(x) \right), x \right) \right) \right] + \lambda \mathcal{W}_{d_z} \left( \bar{f}_e^{*1} \# \mathbb{P}_x, \mathbb{P}_{c^{*1}, \pi^{*1}} \right).
\end{aligned}$$

Noting that $\bar{f}_e \# \mathbb{P}_x = \mathbb{P}_{c, \pi}$ and $Q_C(\bar{f}_e(x)) = \bar{f}_e(x)$, we arrive at

$$\begin{aligned}
&\mathbb{E}_{x \sim \mathbb{P}_x} \left[ d_x \left( f_d \left( \bar{f}_e(x) \right), x \right) \right] \\
&\geq \mathbb{E}_{x \sim \mathbb{P}_x} \left[ d_x \left( f_d^{*1} \left( Q_{C^{*1}} \left( \bar{f}_e^{*1}(x) \right) \right), x \right) \right] + \lambda \mathcal{W}_{d_z} \left( \bar{f}_e^{*1} \# \mathbb{P}_x, \mathbb{P}_{c^{*1}, \pi^{*1}} \right). \tag{18}
\end{aligned}$$

Combining the inequalities in (17) and (18), we obtain Inequality (16) as

$$\mathbb{E}_{x \sim \mathbb{P}_x} \left[ d_x \left( f_d^{*2} \left( \bar{f}_e^{*2}(x) \right), x \right) \right] \leq \mathbb{E}_{x \sim \mathbb{P}_x} \left[ d_x \left( f_d \left( \bar{f}_e(x) \right), x \right) \right]. \tag{19}$$

This concludes our proof.

**Corollary A.1.** *(Corollary 3.1 in the main paper) Consider minimizing the second term: $\min_{f_e, C} \mathcal{W}_{d_z} \left( f_e \# \mathbb{P}_x, \mathbb{P}_{c, \pi} \right)$ in (15) given $\pi$ and assume $K < N$, its optimal solution $f_e^*$ and $C^*$ are also the optimal solution of the following OP:*

$$\min_{f_e, C} \min_{\sigma \in \Sigma_\pi} \sum_{n=1}^{N} d_z \left( f_e(x_n), c_{\sigma(n)} \right), \tag{20}$$

*where $\Sigma_\pi$ is the set of assignment functions $\sigma : \{1, ..., N\} \to \{1, ..., K\}$ such that the cardinalities $\left| \sigma^{-1}(k) \right|, k = 1, ..., K$ are proportional to $\pi_k, k = 1, ..., K$.*

**Proof of Corollary A.1**.

By the Monge definition, we have

$$\mathcal{W}_{d_z}\left(f_e\#\mathbb{P}_x, \mathbb{P}_{c,\pi}\right) = \mathcal{W}_{d_z}\left(\frac{1}{N}\sum_{n=1}^{N}\delta_{f_e(x_n)}, \sum_{k=1}^{K}\pi_k\delta_{c_k}\right) = \min_{T:T\#(f_e\#\mathbb{P}_x)=\mathbb{P}_{c,\pi}}\mathbb{E}_{z\sim f_e\#\mathbb{P}_x}\left[d_z\left(z, T\left(z\right)\right)\right]$$

$$= \frac{1}{N}\min_{T:T\#(f_e\#\mathbb{P}_x)=\mathbb{P}_{c,\pi}}\sum_{n=1}^{N}d_z\left(f_e\left(x_n\right), T\left(f_e\left(x_n\right)\right)\right).$$

Since $T\#\left(f_e\#\mathbb{P}_x\right) = \mathbb{P}_{c,\pi}$, $T\left(f_e\left(x_n\right)\right) = c_k$ for some $k$. Additionally, $\left|T^{-1}\left(c_k\right)\right|, k = 1, ..., K$ are proportional to $\pi_k, k = 1, ..., K$. Denote $\sigma : \{1, ..., N\} \to \{1, ..., K\}$ such that $T\left(f_e\left(x_n\right)\right) = c_{\sigma(n)}, \forall i = 1, ..., N$, we have $\sigma \in \Sigma_\pi$. It also follows that

$$\mathcal{W}_{d_z}\left(\frac{1}{N}\sum_{n=1}^{N}\delta_{f_e(x_n)}, \sum_{k=1}^{K}\pi_k\delta_{c_k}\right) = \frac{1}{N}\min_{\sigma\in\Sigma_\pi}\sum_{n=1}^{N}d_z\left(f_e\left(x_n\right), c_{\sigma(n)}\right).$$

## B    VISUALIZATION OF RECONSTRUCTION RESULTS

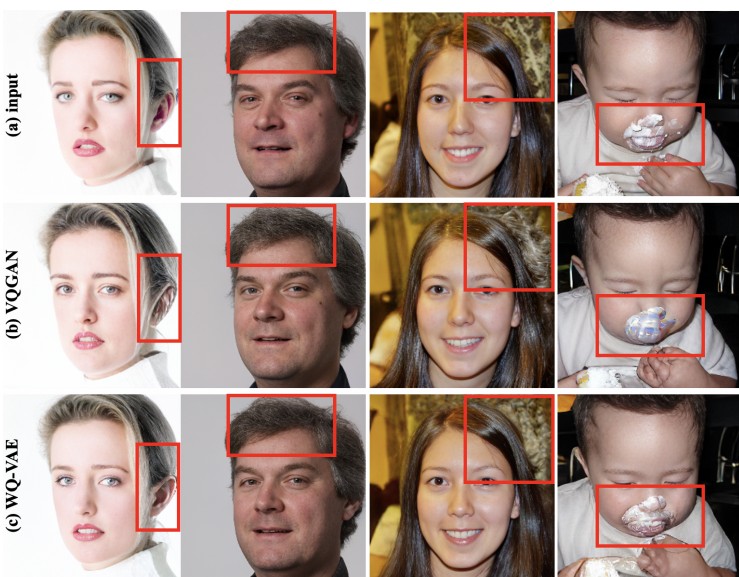

Figure 4: Reconstruction results for the FFHQ dataset.

**Qualitative assessment**: We present the reconstructed samples from FFHQ (high-resolution images) for qualitative evaluation. It can be clearly seen that the high-level semantic features of the input image and colors are better preserved with VQ-WAE than the baseline. Particularly, we notice that VQGAN often produces repeated artifact patterns in image synthesis (see the hair of man is second column in Figure 4) while VQ-WAE does not. This is because VQ-GAN is lack of diversity in the codebook, which will be further analyzed in Section 5.2.1. Consequently, the quantization operator embeds similar patches into the same quantization index and ignores the variance in these patches (e.g., VQ-GAN reconstructs the background in third column of Figure 4 as hair of woman).

## C    EXPERIMENTAL SETTINGS

### C.1    VQ-MODEL

**Implementation**: For fair comparison, we utilize the same framework architecture and hyperparameters for both VQ-VAE and VQ-WAE. Specifically, we construct the VQ-VAE and VQ-WAE models as follows:

- For CIFAR10, MNIST and SVHN datasets, the models have an encoder with two convolutional layers of stride 2 and filter size of $4 \times 4$ with ReLU activation, followed by 2 residual blocks, which contained a $3 \times 3$, stride 1 convolutional layer with ReLU activation followed by a $1 \times 1$ convolution. The decoder was similar, with two of these residual blocks followed by two deconvolutional layers.

- For CelebA dataset, the models have an encoder with two convolutional layers of stride 2 and filter size of $4 \times 4$ with ReLU activation, followed by 6 residual blocks, which contained a $3 \times 3$, stride 1 convolutional layer with ReLU activation followed by a $1 \times 1$ convolution. The decoder was similar, with two of these residual blocks followed by two deconvolutional layers.

- For high-quality image dataset FFHQ, we utilize the well-known VQGAN framework Esser et al. (2021) as the baseline.

We only replace the regularization module of VQ-VAE i.e., two last terms of objective function: $d_z \left( \mathbf{sg} \left( f_e \left( x \right) \right), C \right) + \beta d_z \left( f_e \left( x \right), \mathbf{sg} \left( C \right) \right)$ by our proposed by Wasserstein regularization $\lambda \mathcal{W}_{d_z} \left( f_e \# \mathbb{P}_x, \mathbb{P}_{c,\pi} \right)$ in Eq. (4) for VQ-WAE. Additional, we employ the POT library (Flamary et al., 2021) to compute WS distance for simplicity. However, our VQ-WAE does not require optimal transport map the from WS distance in (6) to update the model. Therefore, we can employ a wide range of speed-up algorithms to solve optimization problem (OP) in (6) such as Sinkhorn algorithm (Cuturi, 2013) or entropic regularized dual for (Genevay et al., 2016).

**Hyper-parameters**: following (Takida et al., 2022), we adopt the adam optimizer for training with: *learning-rate* is $e^{-4}$, *batch size* of 32, *embedding dimension* of 64 and codebook size $|C| = 512$ for all datasets except FFHQ with *embedding dimension* of 256 and $|C| = 1024$. Finally, we train model for CIFAR10, MNIST, SVHN, FFHQ in 100 epoches and for CelebA in 70 epoches respectively.

## C.2 GENERATION MODEL

**Implementation**: It is worth to noting that we employ the codebooks learned from reported VQ-models to extract codeword indices and we employ PixelCNN (Van den Oord et al., 2016) with the same setting for generation for all VQ-VAE, SQ-VAE and VQ-WAE. In particular, we feed PixelCNN over the "pixel" values of the $8 \times 8$ 1-channel latent space for CIAR10, MNIST, SVHN, and $16 \times 16$ 1-channel latent space for CelebA.

**Hyper-parameters**: we adopt the adam optimizer for training with: *learning-rate* is $3e^{-4}$, *batch size* of 32. Finally, we PixelCNN over the "pixel" values of the $8 \times 8$ 1-channel latent space in 100 epoches.

