# OpenReview forum: "Vector Quantized Wasserstein Auto-Encoder"
_ICLR.cc/2023/Conference — Submitted to ICLR 2023_

### Official Review · Reviewer_7LM4 · 2022-10-24

**Confidence:** 5
**Correctness:** 1
**Technical Novelty And Significance:** 2
**Empirical Novelty And Significance:** 2
**Recommendation:** 3

**Clarity, Quality, Novelty And Reproducibility:**

The paper is well-written in general. The novelty is kind of incremental, even if the above-mentioned weakness (1) is corrected. The reproducibility is satisfactory as the code is available.

**Strength And Weaknesses:**

Strength:

(1) The paper is well-written in general.

(2) The empirical experimental results seem to be convincing.

Weaknesses:

(1) The theoretical derivations are not sound, because, in both the VQ-VAE and the presented VQ-WAE, many (like 8 $\times$ 8) latent codes correspond to one input image $x$, whereas the presented theorems are derived based on one latent code per x. In other words, there is a mismatch between theory and implementation.

(2) No explicit demonstrations like the reconstructed images are shown, which makes it difficult to evaluate the quality of the vector quantization. For example, is the reconstruction quality better than that of the VQ-GAN?

**Summary Of The Paper:**

A new vector quantization method is proposed, named Vector Quantized Wasserstein Auto-Encoder (VQ-WAE). The presented VQ-WAE employs the Wasserstein (WS) distance in both the observation x space and the latent z space to encourage matching, mimicking the existing Wasserstein Auto-Encoder. Experiments on MNIST, CIFAR10, SVHN, and CelebA datasets are conducted to demonstrate the superiority of the VQ-WAE to the existing VQ-VAE and SQ-VAE methods.

**Summary Of The Review:**

The main weakness of the proposed VQ-WAE lies in its mismatch between theory and implementation. Besides, it's not clear if the presented VQ-WAE works indeed better than existing VQ methods like the VQ-GAN.

---

> ### Author Response · Authors · 2022-11-15
> **Response to Reviewer 7LM4  (1/2)**
>
> Thank you very much for your time and effort for this review. We have carefully addressed your comments in the following:
>
> &nbsp;
>
> ***Comment:* The theoretical derivations are not sound, because, in both the VQ-VAE and the presented VQ-WAE, many (like $8 \times 8$) latent codes correspond to one input image , whereas the presented theorems are derived based on one latent code per x. In other words, there is a mismatch between theory and implementation.**
>
> *Answer:*
>
> Thanks for this comment. Traditionally, to simplify the context, VQ-based works for learning discrete representations [Van Den Oord et al., 2017; Razavi et al., 2019; Roy et al., 2018; Takida et al., 2022] usually present their derivations and frameworks based on one latent code per sample $x$, while using $N\times N$ (e.g., ($8 \times 8$)) latent codes correspond to one input image in the implementation.
>
> We follow this convention  because the theory and technicality developed in our paper are quite complicated and we wish to present the main ideas and spirit of our approach. Moreover,  our algorithm still works with $N\times N$ latent codes correspond to one input image as shown in our released implementation.
>
> This comment is very critical. We really appreciate if the reviewer reexamines our theory and technicality.
>
> &nbsp;
>
> ***Comment:* No explicit demonstrations like the reconstructed images are shown, which makes it difficult to evaluate the quality of the vector quantization. For example, is the reconstruction quality better than that of the VQ-GAN?**
>
> *Answer:* Comparison with VQ-GAN:
>
> **Quantitative assessment:** We additionally conduct experiment on high-quality image dataset FFHQ, we apply VQ-WAE to the well-known VQGAN framework to demonstrate the applicability of our proposed method on large dataset and framework. We recap the results in comparison with VQ-GAN from the Table 1 in the main paper:
>
> |Dataset | Model | Latent Size | SSIM $\uparrow$ | PSNR $\uparrow$ | LPIPS $\downarrow$ | rFID$ \downarrow$ | Perplexity $\uparrow$ |
> | :---- | :----: | :----: | ----: | ----: | ----: | ----: | ----: |
> |FFHQ | VQGAN | 16 $\times$ 16 | 0.6641 | 22.24 | **0.1175** | 4.42 | 423 |
> | | VQ-WAE | 16 $\times$ 16 | **0.6648** | **22.45** | 0.1245 | **4.20** | **1022** |
>
>
> The results in Table 1 show that VQ-WAE  outperform VQ-GAN on most metrics. Especially, the result of rFID (which evaluate the image quality at the dataset level) confirm that the sampled images from VQ-WAE are of better quality than these from VQ-GAN, which is in line with the qualitative results shown below.
>
> **Qualitative assessment**: We present the reconstructed samples from  FFHQ (high-resolution images) for qualitative evaluation in Figure 4, Appendix B. It can be clearly seen that the high-level semantic features of the input image and colors are better preserved with VQ-WAE than the baseline. Particularly, we notice that VQGAN often produces repeated artifact patterns in image synthesis (see the hair of man is second column in Figure 4 (Appendix.B) while VQ-WAE does not. This is because VQ-GAN is lack of diversity in the codebook.
> Consequently, the quantization operator embeds similar patches into the same quantization index and ignores the variance in these patches (e.g., VQ-GAN reconstructs the background in third column of Figure 4 as hair of woman).

---

> > ### Author Response · Authors · 2022-11-15
> > **Response to Reviewer 7LM4  (2/2)**
> >
> > ***Comment on Novelty:* The presented VQ-WAE employs the Wasserstein (WS) distance in both the observation x space and the latent z space to encourage matching, mimicking the existing Wasserstein Auto-Encoder (WAE)**
> >
> > *Answer:*
> >
> > We agree that our work is inspired from WAE as we state clearly in our paper. However, we solve the learning discrete representation problem, while WAE solves the learning continuous representation problem. Two problems are distinctive, hence our theoretical development is distinctively different from that in WAE. Moreover, our theory development naturally leads to using the WS distance $\\min_{f_e,C} \\mathcal{W}_{d_z} \\left(f_e \\# \\mathbb{P}_x,\\mathbb{P}_c,_\pi \\right)$  which has a strong impact on learning discrete representations for more controllable codewords as we demonstrate in Corolarry 3.1.
> >
> > More importantly, our proof is totally different from that in WAE. Therefore, we strongly believe our theoretical developments are novel. To further clarify, we brief the pathway of our theory development as follows:
> >
> > Starting with OP (1) [note that $\mathbb{X}\in \mathbb{R}^V$ while $C \in \mathbb{R}^D$ where $D << V$:]:
> >
> > - $$
> > \\min_{C,\pi} \\min_{f^d} \\mathcal{W}\_{d\_x} \\left( f_d \\# \\mathbb{P}_c,_\pi ,\\mathbb{P}_x \\right)
> > $$
> >
> > In OP (1) we propose to learn a *"deterministic decoder transporting the codeword to data distributions"* via minimizing the WS distance between them (i.e., finding transport map between two random variables living in the data space $\mathcal{X}$).
> >
> >
> > However, it's hard to optimize WS distance on  high dimensional data space $\mathcal{X}$. Theorem 3.1 engages the OP (1) with the latent space which results the OP (3) (i.e., finding a transport map between two random variables living in the latent space $\mathcal{Z}$):
> >
> > - $$
> > \\min_{C,\pi,f_d} \\min_{\\bar{f}_e : \\bar{f}_e \\# \\mathbb{P}_x=\mathbb{P}{c,\pi}} \\mathbb{E}\_{x \\sim \mathbb{P}\_x}
> > \\left[
> > d_x \\left( f_d \\left( \\bar{f}_e \\left( x \\right) \\right),x \\right)
> > \\right]
> > $$
> >
> > More specifically, we turn the WS minimization in OP (1) to *"push-forwarding the data to codeword distribution"* via *"minimizing a WS distance between the latent representation and codeword distributions"*. (the underline equation in second **min**).
> >
> > Since $\bar{f}_e$ is a deterministic encoder, there is no real gradient defined. To make it trainable,
> > we replace $\bar{f}_e$ by a continuous encoder $f_e: \\mathcal{X} \\rightarrow \\mathcal{Z}$
> > and arrive at the OP (4):
> >
> > - $$
> > \min_{C,\pi}\min_{f_d,f_e}  \\left\\{
> > \\mathbb{E}_{x \\sim \\mathbb{P}_x}  \\left[d_x \\left( f_d \\left(Q \\left( f_e\\left( x \\right) \\right) \\right), x \\right) \\right] +
> > \\lambda \\mathcal{W}\_{d\_z} \\left( f_e  \\# \\mathbb{P}_x, \\mathbb{P}_c,_\pi  \\right)
> > \\right\\}
> > $$
> >
> > Finally, our Theorem 3.2 show that (1), (3) and (4) are equivalent. To summarize, starting from (1) formulated on high-dimension data space, we develop sufficient and novel theories to turn it to learning on latent space in (3) and (4).
> >
> > &nbsp;
> >
> > **The mismatch between theory and implementation.**
> >
> > We conjecture that you are mentioning to the mismatch between learning $\pi$ in OP (4) and fixing $\pi$ to the uniform distribution in Algorithm 1.
> >
> > Although there is a relaxation in learning or fixing $\pi$, this is theoretically justified in Corollary 3.1. Particularly, our Corollary 3.1 says that the proportion of latent representations assigned to a codeword $c_k$ is proportional to $\pi_k$ or equivalently *the cardinalities $\left|(\sigma^\*)^{-1}\left(k\right)\right|,k=1,...,K$
> > are proportional to $\pi_{k},k=1,...,K$*.
> >
> > In the context of learning discrete representations, it is a desirable behaviour by making latent representations more controllable because we want latent representations equally are assigned to codewords to maximize the codeword utilization.
> >
> > Therefore, in our empirical algorithm, we actively fix  $\pi$ as the uniform distribution to maximize the codeword utilization and perplexity.

---

> ### Author Response · Authors · 2022-11-27
> **Looking forward to your reply!**
>
> Dear reviewers,
>
> We first thank you again for your valuable comments and detailed suggestions. In the previous replies, we have tried our best to address your questions and revised the manuscript based on the suggestions.
>
> We are looking forward to your reply to our responses, and we are open to any discussions to improve this work.
>
> Best wishes!

---

### Official Review · Reviewer_zbTA · 2022-10-24

**Confidence:** 5
**Correctness:** 4
**Technical Novelty And Significance:** 2
**Empirical Novelty And Significance:** Not applicable
**Recommendation:** 5

**Clarity, Quality, Novelty And Reproducibility:**

* The paper could be reorganized and rewritten to increase clarity and conciseness.

* The proposed approach lacks novelty and originality. Moreover, the various choices, like dropping the stop gradient or assuming that the codewords have uniform weights, are not very well justified in the paper.





**Strength And Weaknesses:**

### Strengths

* The idea of using Wasserstein distance for vector quantization is interesting.

### Weaknesses

* The paper is not well-written, and the flow of the paper can significantly benefit from a significant rewriting.

* The paper's novelty is limited to applying the Wasserstein distance in the latent space of an autoencoder with a moving sparse target distribution (i.e., the codeword distribution).

* Given the deterministic nature of both the encoder and decoder and applying the Wasserstein distance in the embedding space, the paper is more closely related to the Sliced-Wasserstein Auto-Encoders (SWAE), Kolouri, et al. 2018, as opposed to the WAE that uses probabilistic encoder/decoders.

* The choice of uniform $\pi$s is not well justified in the paper. Also, the authors may want to use a different symbol for $\pi$, because $\pi$ is often used to denote the `transport plan' in the optimal transport-related literature and in calculation of the Wasserstein distance.

* The stop gradient used in VQ-VAE could also be used in your formulation. In particular, if you freeze the encoder, the optimal codeword could be directly obtained from the barycentric projection of the transportation plan. Some discussions on these aspects of the training could improve the quality of the paper.

* Eq (3) is missing a $f_d$ for the inner minimization.

* The paragraphs following Eq (3) state $\bar{f}_e$ is deterministic repeatedly. There are several instances of repeated reworded statements throughout the paper, which flags that the paper could use significant polishing to improve the flow and communicate the concepts more concisely.






**Summary Of The Paper:**

The paper proposes a vector quantized autoencoder that minimizes the Wasserstein distance between the encoded samples and the code word. In short, the loss function is simply the quantized autoencoding loss plus a penalty term that measures the Wasserstein distance between the empirical distribution of the encoded samples and the empirical distribution of the code words. The authors then evaluate their approach on the generative modeling for CIFAR10, MNIST, SVHN, and CelebA datasets showing competitive performance compared to VQ-VAE and SQ-VAEs.

**Summary Of The Review:**

While the paper has exciting aspects, I think its current state lacks sufficient quality for ICLR. In particular, the paper can benefit from reorganization. Also, the authors could provide a finer prior work section that better places their work among the existing approaches. Finally, detailed discussions on various choices throughout the paper could benefit this paper. To summarize, I think this paper has potential, however, it is not ready for publication yet.

---

> ### Author Response · Authors · 2022-11-15
> **Response to Reviewer zbTA  (1/2)**
>
> Thank you very much for your time and effort for this review. We have carefully addressed your comments in the following:
>
> &nbsp;
>
> ***Comment the paper's flow and Novelty:***
>
> - **The paper is not well-written, and the flow of the paper can significantly benefit from a significant rewriting.**
>
> - **The paper's novelty is limited to applying the Wasserstein distance in the latent space of an autoencoder with a moving sparse target distribution (i.e., the codeword distribution).**
>
> *Answer:*
>
> We strongly believe our theoretical developments are novel and the flow of theoretical development is easy to follow as commented by other reviewers. To further clarify, we brief the pathway of our theory development which are presented in the paper as follows:
>
> Starting with OP (1) [note that $\mathbb{X}\in \mathbb{R}^V$ while $C \in \mathbb{R}^D$ where $D << V$:]:
>
> - $$
> \\min_{C,\pi} \\min_{f^d} \\mathcal{W}\_{d\_x} \\left( f_d \\# \\mathbb{P}_c,_\pi ,\\mathbb{P}_x \\right)
> $$
>
> In OP (1) we propose to learn a *"deterministic decoder transporting the codeword to data distributions"* via minimizing the WS distance between them (i.e., finding transport map between two random variables living in the data space $\mathcal{X}$).
>
>
> However, it's hard to optimize WS distance on  high dimensional data space $\mathcal{X}$. Theorem 3.1 engages the OP (1) with the latent space which results the OP (3) (i.e., finding a transport map between two random variables living in the latent space $\mathcal{Z}$):
>
> - $$
> \\min_{C,\pi,f_d} \\min_{\\bar{f}_e : \\bar{f}_e \\# \\mathbb{P}_x=\mathbb{P}{c,\pi}} \\mathbb{E}\_{x \\sim \mathbb{P}\_x}
> \\left[
> d_x \\left( f_d \\left( \\bar{f}_e \\left( x \\right) \\right),x \\right)
> \\right]
> $$
>
> More specifically, we turn the WS minimization in OP (1) to *"push-forwarding the data to codeword distribution"* via *"minimizing a WS distance between the latent representation and codeword distributions"*. (the underline equation in second **min**).
>
> Since $\bar{f}_e$ is a deterministic encoder, there is no real gradient defined. To make it trainable,
> we replace $\bar{f}_e$ by a continuous encoder $f_e: \\mathcal{X} \\rightarrow \\mathcal{Z}$
> and arrive at the OP (4):
>
> - $$
> \min_{C,\pi}\min_{f_d,f_e}  \\left\\{
> \\mathbb{E}_{x \\sim \\mathbb{P}_x}  \\left[d_x \\left( f_d \\left(Q \\left( f_e\\left( x \\right) \\right) \\right), x \\right) \\right] +
> \\lambda \\mathcal{W}\_{d\_z} \\left( f_e  \\# \\mathbb{P}_x, \\mathbb{P}_c,_\pi  \\right)
> \\right\\}
> $$
>
> Finally, our Theorem 3.2 show that (1), (3) and (4) are equivalent. To summarize, starting from (1) formulated on high-dimension data space, we develop sufficient and novel theories to turn it to learning on latent space in (3) and (4).
>
> &nbsp;
>
> ***Comment:* Given the deterministic nature of both the encoder and decoder and applying the WS distance in the embedding space, the paper is more closely related to the SWAE as opposed to the WAE that uses probabilistic encoder/decoders.**
>
> *Answer:*
>
> SWAE proposed to add the sliced-Wasserstein distance between the distribution of the encoded training samples and a samplable prior distribution to the auto encoder (AE) loss. The difference compared to the WAE lies in using the usage of the sliced-Wasserstein distance instead of GAN or MMD-based penalties.
>
> However, both WAE and SWAE perform on continuous latent representation regardness of probabilistic encoder/decoders or deterministic encoder/decoders. Therefore, they totally different from our VQ-WAE or other vector quantized-based autoencoder models which work on discrete latent representation. VQ-WAE seeks a \emph{deterministic discrete} encoder $\bar{f}_{e}$ mapping data examples
> directly to discrete codewords, concurring with vector quantization and serving our further derivations. More importantly, our proof is totally different from that in WAE.
>
> &nbsp;
>
> ***Comment:*
> The choice of uniform s is not well justified in the paper.**
>
>
> *Answer:*
>
> We believe this choice is theoretically justified by our Corollary 3.1. Particularly, our Corollary 3.1 says that the proportion of latent representations assigned to a codeword $c_k$ is proportional to $\pi_k$ or equivalently *the cardinalities $\left|(\sigma^\*)^{-1}\left(k\right)\right|,k=1,...,K$
> are proportional to $\pi_{k},k=1,...,K$*.
>
> In the context of learning discrete representations, it is a desirable behaviour by making latent representations more controllable because we want latent representations equally are assigned to codewords to maximize the codeword utilization.
>
> Therefore, in our empirical algorithm, we actively fix  $\pi$ as the uniform distribution to maximize the codeword utilization and perplexity.

---

> > ### Author Response · Authors · 2022-11-15
> > **Response to Reviewer zbTA  (2/2)**
> >
> > ***Comment:* The stop gradient used in VQ-VAE could also be used in your formulation. Some discussions on these aspects of the training could improve the quality of the paper.**
> >
> > *Answer:*
> >
> > In the VQ-VAE, $f_e(x)$ is the output of the encoder network, and $C$ is the embedding. They are mutually-related, and both need to be optimized. Stop-gradients separates the loss to the **codebook alignment loss** and the **commitment loss** as an Alternating Projections kind of optimization algorithm, instead of optimizing two mutually-related subsystems simultaneously, it "freezes" one while optimizes the other, so that the optimization will not "collapse" into a trivial wrong solution.
> >
> > Particularly in VQ-VAE paper, this is close similarity to k-means algorithm, by alternating between (phase 1) estimating centroids and (phase 2) deciding which element belongs to each centroid. You can see that equations (1) and (2) in the paper are essentially a k-means criterion [ref. Appendix A.1 (Van Den Oord et al., 2017)]. The paper also emphasized that the stop-gradient in different terms of the loss will cause the loss to effect learning in different subsystems of the overall system.
> >
> > By contrast, VQ-WAE can simultaneously optimize f(e) and $C$, and guarantee that $C$ become the optimal clustering centroids of the optimal clustering solution which minimizes the distortion (ref. our Lemma 3.1). Therefore, we do not applied stop gradient in our formulation.
> >
> > We also conduct additionally conduct experiments with stop-gradient and obtain the similar performance.
> >
> > &nbsp;
> >
> > ***Comment:* Eq (3) is missing a  for the inner minimization.**
> >
> > *Answer:*
> >
> > Thank you for the correction, we have updated the Eq (3) in the revised version.

---

> ### Author Response · Authors · 2022-11-27
> **Looking forward to your reply**
>
> Dear reviewers,
>
> We first thank you again for your valuable comments and detailed suggestions. In the previous replies, we have tried our best to address your questions and revised the manuscript based on the suggestions.
>
> We are looking forward to your reply to our responses, and we are open to any discussions to improve this work.
>
> Best wishes!

---

### Official Review · Reviewer_9V99 · 2022-10-24

**Confidence:** 5
**Correctness:** 2
**Technical Novelty And Significance:** 2
**Empirical Novelty And Significance:** 2
**Recommendation:** 3

**Clarity, Quality, Novelty And Reproducibility:**

## Clarity

As mentioned, it is unclear how Algorithm 1 is implemented with backpropergation.

## Quality

This paper is written in plain Ensligh and easy to follow most of the time. In Table 2, MSE is mentioned in the title but not included in the table.

## Novelty

Theorem 3.1 is a classical result in $K$-means clustering and vector quantization. Theorem 3.2 is incorrect.

## Reproducible

Code is attached for reproducibility.


**Strength And Weaknesses:**

## Strengths

Application of the WAE framework to discrete representation learning is reasonably demonstrated with various evaluation metrics. The derivation of objective function (4) from the Wasserstein distance minimization problem (1) is natural at least intuitively.


## Weaknesses

1. The major weakness is the gap between the theory and algorithm. The optimization problems (1), (2) and (4), which are claimed to be equivalent to each other in Theorem 3.2, are basically about fitting a $K$-support discrete latent distribution $\sum_{k=1}^K \pi_k \delta_{c_k}$ to the data distribution by minimizing the Wasserstein distance between them. In particular, the mass $\pi_k$ assigned to cordwood $c_k$ is an optimization variable to be fitted. However, in the actual algorithm (Algorithm 1), it is fixed as $1/K$ and only $c_k$'s are fitted. So Algorithm 1 in fact solves a different problem from (4). This distinction is important since the so-called "codebook collapse" occurs precisely because some $\pi_k$ is zero but is destined to be prevented if the proportion of codewords is forced to be uniform. In fact, that the optimal transport problem (1) is equivalent to the $K$-means clustering has been known at least since 1982 (Pollard, D., 1982. Quantization and the method of k-means. IEEE Transactions on Information theory, 28(2), pp.199-205). In light of this result, the modified problem that Algorithm 1 solves is the $K$-means clustering problem under the constraint that each (latent) cluster contains the same number of sample (latent) points. I do not think that this modified problem is necessarily better or more desirable than the original $K$-means problem (1). Suppose (1) is solved exactly and there is a codebook collapse. Then this means that strictly less than $K$ codewords best represent the data. Is this bad? So the case that codebook collapse becomes problematic is when the algorithm is trapped in a bad local minimum. This is a problem of the algorithm, rather than the sin of codebook underutilization. In this sense, the role of the added constraint is to *regularize* the algorithm to avoid bad local minima. On the other hand, Takida et al. (2022) avoids the same problem by using stochastic quantization but not constraining cluster sizes.

2. The proof of Theorem 3.2 is incomplete. The proof only states that the (3) $\leq$ (4). The possibility of strict inequality cannot be ruled out because the constructed variables $(C^{*2}, \pi^{*2}, f_d^{*2}, \bar{f}_e^{*2})$ are only feasible for (3) but not necessarily optimal. Further, measurability of $\bar{f}_e^{*2}$ needs to be proved.

3. The objective of Algorithm 1 is not differentiable because of the $\arg\min$ operator in Line 5 and the Wasserstein distance penalty (6).  How this is put to the gradient learning pipeline is not stated. Is the  gradient passing trick due to van den Oord et al. (2017) used to deal with the $\arg\min$ operator? Or the Gumbel-softmax trick? is (6) smoothed with entropy regularization? If the answer to the first and the third questions are yes (which I presume), then the real contribution of this paper appears to be replacing the last two terms in the VQ-VAE objective with (a smoothed version of) (6); see Sect. B.1.

4. Finally, I wonder what are the authors' opinion about some peaks of codewords in Fig. 2a when $|C|$ is large in the MNIST dataset.


**Summary Of The Paper:**

This paper extends the Wasserstein autoencoder (WAE) framework due to Tolstikhin et al. (2017) to incorporate discrete representations as considered by the vector quantized variational autoencoder (VQ-VAE) due to van den Oord et al. (2017). It begins with a theoretical development leading to the optimization problem (4) and then propose an approximate solution procedure in Algorithm 1. The performance of the proposed algorithm is tested on the standard benchmark, namely CIFAR10, MNIST, SVHN, and CelebA. Comparisons are made with VQ-VAE and SQ-VAE (Takida et al., 2022), a stochastic variant of the former. While for the traditional pixel- and patch-level metrics SQ-VAE exhibits better performances, VQ-WAE has merits in dataset-level metrics such as rFID, and Shannon entropy of the latent codeword distribution. The latter phenomenon can be understood as that VQ-WAE yields better codebook utilization and thus prevents codebook collapse problem.

**Summary Of The Review:**

There are gap between the theory developed and the actual algorithm, both of which have unresolved flaws.

---

> ### Author Response · Authors · 2022-11-15
> **Response to Reviewer 9V99 (1/3)**
>
> Thank you very much for your time and effort for this review. We have carefully addressed your comments in the following:
>
> &nbsp;
>
> We would like to break down the first comment of the reviewer which concerns two main points:
>
> - The novelty of theoretical development, particularly Theorem 3.1.
> - The gap between theory and algorithm.
>
> and address each of them as follows:
>
> &nbsp;
>
> ***Comment 1-1:* The optimization problems (1), (2) and (4), which are claimed to be equivalent to each other in Theorem 3.2, are basically about "fitting a support discrete latent distribution to the data distribution by minimizing the Wasserstein distance between them".**
>
> *Answer:*
>
> First of all, we believe our theory development for using Wasserstein distance to learn discrete representations is solid and novel. To further clarify, we brief the pathway of our theory development as follows:
>
> Starting with OP (1) [note that $\mathbb{X}\in \mathbb{R}^V$ while $C \in \mathbb{R}^D$ where $D << V$:]:
>
>
> - $$
> \\min_{C,\pi} \\min_{f^d} \\mathcal{W}\_{d\_x} \\left( f_d \\# \\mathbb{P}_c,_\pi ,\\mathbb{P}_x \\right)
> $$
>
> In OP (1) we propose to learn a *"deterministic decoder transporting the codeword to data distributions"* via minimizing the WS distance between them (i.e., finding transport map between two random variables living in the data space $\mathcal{X}$).
>
>
> However, it's hard to optimize WS distance on  high dimensional data space $\mathcal{X}$. Theorem 3.1 engages the OP (1) with the latent space which results the OP (3) (i.e., finding a transport map between two random variables living in the latent space $\mathcal{Z}$):
>
> - $$
> \\min_{C,\pi,f_d} \\min_{\\bar{f}_e : \\bar{f}_e \\# \\mathbb{P}_x=\mathbb{P}{c,\pi}} \\mathbb{E}\_{x \\sim \mathbb{P}\_x}
> \\left[
> d_x \\left( f_d \\left( \\bar{f}_e \\left( x \\right) \\right),x \\right)
> \\right]
> $$
>
> More specifically, we turn the WS minimization in OP (1) to *"push-forwarding the data to codeword distribution"* via *"minimizing a WS distance between the latent representation and codeword distributions"*. (the underline equation in second **min**).
>
> Since $\bar{f}_e$ is a deterministic encoder, there is no real gradient defined. To make it trainable,
> we replace $\bar{f}_e$ by a continuous encoder $f_e: \\mathcal{X} \\rightarrow \\mathcal{Z}$
> and arrive at the OP (4):
>
> - $$
> \min_{C,\pi}\min_{f_d,f_e}  \\left\\{
> \\mathbb{E}_{x \\sim \\mathbb{P}_x}  \\left[d_x \\left( f_d \\left(Q \\left( f_e\\left( x \\right) \\right) \\right), x \\right) \\right] +
> \\lambda \\mathcal{W}\_{d\_z} \\left( f_e  \\# \\mathbb{P}_x, \\mathbb{P}_c,_\pi  \\right)
> \\right\\}
> $$
>
> Finally, our Theorem 3.2 show that (1), (3) and (4) are equivalent. To summarize, starting from (1) formulated on high-dimension data space, we develop sufficient and novel theories to turn it to learning on latent space in (3) and (4).
>
> &nbsp;
>
> ***Comment on Novelty:* Theorem 3.1 is a classical result in k-means clustering and vector quantization.**
>
> *Answer:*
>
> We strongly believe that Theorem 3.1 is novel and different from classical result in k-means clustering and vector quantization.
>
> Specifically, given $\mathbb{X}\in \mathbb{R}^V$, the problem of k-level quantizer [Pollard, D., 1982] searches for k discrete latent $\{a_1,...,a_k\}$ in the same dimension (i.e. $a_i \in \mathbb{R}^V$) while
> in our problem, we search $C \in \mathbb{R}^D$ where $D << V$. Therefore, the results in [Pollard, D., 1982] are more related to our Lemma 3.1 (OP (2)) which is used to offer more intuition for the OP (1) rather than Theorem 3.1 as which optimizes the optimal transport on latent space as in OPs (3) and (4) [please refer our answer in in Comment 1-1].

---

> > ### Author Response · Authors · 2022-11-15
> > **Response to Reviewer 9V99 (2/3)**
> >
> > ***Comment on the gap between the theory and algorithm:* The mass $\pi_k$ assigned to codeword $c_k$ is an optimization variable to be fitted. However, in the actual algorithm (Algorithm 1), it is fixed as $\frac{1}{K}$ and only $c_k$'s are fitted. So Algorithm 1 in fact solves a different problem from (4).**
> >
> > *Answer:*
> >
> > In the OP (4), WS term  $\\mathcal{W}_{d_z} \left(f_e \\# \\mathbb{P}_x,\\mathbb{P}_c,_\pi \right)$  operates on latent space where output of the encoder network $f_e:\\mathcal{X} \\rightarrow \\mathcal{Z}$, $\pi$ and the codebook $C$ are jointly optimized.
> >
> > We agree that Algorithm 1 solves a slight different problem form (4) because in Algorithm 1, we fix $\pi$ as the uniform distribution and do not learn $\pi$.
> >
> > However, as guided by our Corollary 3.1, we have a strong reason to do this. Particularly, our Corollary 3.1 says that the proportion of latent representations assigned to a codeword $c_k$ is proportional to $\pi_k$ or equivalently *the cardinalities $\left|(\\sigma^\*)^{-1}\left(k\right)\right|,k=1,...,K$ are proportional to $\\pi_k,k=1,...,K$.*
> >
> > In the context of learning discrete representations, it is a desirable behaviour by making latent representations more controllable because we want latent representations equally are assigned to codewords to maximize the codeword utilization.
> >
> > Therefore, in our empirical algorithm, we actively fix  $\pi$ as the uniform distribution to maximize the codeword utilization and perplexity.
> >
> > &nbsp;
> >
> > ***Comment 1-2:* The optimal transport problem (1) is equivalent to the K-means clustering [Pollard, D., 1982. Quantization and the method of k-means]**
> >
> > *Answer:*
> >
> > We agree with the reviewer that OP (1) and is equivalent to the K-means clustering as they hold same optimal solution (see our Lemma 1). However, solving (1) or K-means clustering directly in high-dimensional data space (e.g., images or texts) is infeasible because (i) the curse of dimensionality and (ii) even we do not have a good way to compute distance between two structural data examples such as images or texts. Additionally, K-menas is a NP-hard problem due to the combinatoric explosion of the clustering assignments. Practical algorithm to solve K-means such as (i) assigning a data point to the closest centroid and (ii) updating a centroid as the mean of data points assigned to it easily gets stuck in a local optimum and strongly depends on centroid initialization.
> >
> > &nbsp;
> >
> > ***Comment 1-3:***
> > - **In light of this result (comment 1-2), Algorithm 1 solves is the k-means clustering problem under the constraint that each (latent) cluster contains the same number of sample (latent) points. I do not think that this modified problem is necessarily better or more desirable than the original k-means problem (1).**
> >
> > - **Suppose (1) is solved exactly and there is a codebook collapse. Then this means that strictly less than  codewords best represent the data. Is this bad?**
> >
> >  *Answer:*
> >
> > We would like to re-note that OP (1), results in [Pollard, D., 1982] or our Lemma 3.1 is optimized on data space while Algorithm 1 is derived form OP (4) which finds optimal transport on latent space.
> >
> > Because learning codewords and clusters on the data space is not efficient, in Theorem 3.1, we equivalently turn it to the latent space and come with OP in (4).
> >
> > In your description of the code collapse problem, we understand it as if latent representations (or data) in nature has unknown $K_{opt}$ clusters characterized by $K_{opt}$ codewords, however, if we search for $K>K_{opt}$ codewords, some codewords become redundant (i.e., its $\pi_k=0$), hence setting $\pi$ as the uniform distribution and forcing latent representations equally assigned to codewords is not a good idea.
> >
> > Here we note that latent representations are learnable and adaptable via the encoder $f_e$. Therefore, both latent representations and codewords are learnable and adjustable so that there is no redundant codewords and latent representations equally assigned to codewords.
> > Paricularly, clustering problem and learning codewords/centroids is an unsupervised problem. *If we can view data/latent representations in $K$ clusters for example, we can also use more clusters and codewords/centroids (e.g., $2K$ clusters and codewords/centroids) to characterise data/latent representations because a cluster can be viewed as a combination of some smaller clusters.* Our theorem guarantees that the codewords will spread over data/latent representations so that for the optimal assignments, the proportion of latent representations/data to codewords is proportional to $\pi_k, k=1,...,K$.
> >
> > Moreover, our experimental results in Table 2 and Figure 2 totally support our claim. This also shows the advantages of solving (4) on the latent space rather than solving (1) on the data space for which data examples are not learnable and adjustable to fit codewords or centroids.

---

> > > ### Author Response · Authors · 2022-11-15
> > > **Response to Reviewer 9V99 (3/3)**
> > >
> > > ***Comment 2:*
> > > The proof of Theorem 3.2 is incomplete. Further, measurability of $\\bar{f}_e^{\*2}$ needs to be proved.**
> > >
> > > *Answer:*
> > >
> > > We appreciate the reviewer for reading our proof and thank you for the detailed and helpful comment. First of all, we have added the proof of measurability of $\bar{f}_{e}^{*2}$ (ref. Lemma A.2 in appendix). Then, along with Lemma A.2, we have updated and modified the proof of Theorem 3.2 for completion and improving its readability. *This is a critical comment and we really appreciate if the reviewer can have another round for our proof*.
> > >
> > > &nbsp;
> > >
> > > ***Comment 3:*
> > > The objective of Algorithm 1 is not differentiable because of the operator in Line 5 and the Wasserstein distance penalty (6).**
> > >
> > > *Answer:*
> > >
> > > We use the gradient trick [van den Oord et al. (2017)] to deal with the back-propagation from decoder to encoder for reconstruction term. For Wasserstein distance penalty, our derivation replace $\bar{f}_e$ by a continuous encoder $f_e: \\mathcal{X} \\rightarrow \\mathcal{Z}$ in OP (4) [ref. theorem 3.2] to make it trainable.
> > > We have clarified it in the revised paper.
> > >
> > > &nbsp;
> > >
> > > ***Comment 4:*
> > > Some peaks of codewords in Fig. 2a when $\|C\|$ is large in the MNIST dataset.**
> > >
> > > *Answer:*
> > >
> > >
> > > Due the imperfection of optimization, the codebook is not perfectly uniform, however, it is very close to the uniform distribution as we expect. Moreover, it's worth noticing the scale of the y-axis. The difference in value and ratio between peaks and general histograms of VQ-WAE are small for all $\|C\|$. Particularly for $\|C\|=512$, peaks are 2500 while general histograms are from 1500 to 2000. In contrast, the differences between histograms of VQ-VAE and SQ-VAE are very large, ranging from several thousand to 50000.

---

> > > ### Comment · Reviewer_9V99 · 2022-11-18
> > > **K-means clustering**
> > >
> > > I think it is helpful to distinguish the K-means clustering *problem* and K-means clustering *algorithm*. The K-means clustering problem is the optimal transport of Pollard (1982) or the K-level vector quantization problem (Gersho and Gray, 1992). Let us call "the practical algorithm to solve K-means such as (i) assigning a data point to the closest centroid and (ii) updating a centroid as the mean of data points assigned to it" by the K-means clustering algorithm. As you mentioned, the K-means clustering problem is NP-hard and the K-means clustering algorithm is suboptimal, especially in high dimensions. There can be other algorithms to approximately solve the K-means clustering problem, one of which is yours that tackles the problem by embedding to a latent space. My point is that, if the K-means clustering problem is perfectly solved by some oracle and some of the optimal $\pi_k$'s turn out to be zero, can you call this result "bad" and should be avoided. In an extreme case, a 2K-means clustering problem may possess only K supports, depending on the data distribution, leaving K of the 2K clusters empty (This is according to your Corollary 3.1). Codebook collapse only matters when the globally optimal solution to the K-means clustering problem has K supports while the algorithm gets stuck at a local minimum with less than K supports. Forcing equal cluster sizes can be a good heuristic to regularize the algorithm, but should not be the gold standard.

---

> > > > ### Author Response · Authors · 2022-12-12
> > > > **Additional discussion**
> > > >
> > > > We agree with the reviewer that "Forcing equal cluster sizes can be a good heuristic to regularize the algorithm, but should not be the gold standard".
> > > >
> > > > We would like to discuss more the case *K-means clustering problem is perfectly solved by some oracle and some of the optimal 's turn out to be zero*:
> > > >
> > > > We argue that K-means clustering only is optimal when all the centroids are active. This concurs with the analysis of Pollard (1982): "if P does not concentrate on k - 1 or fewer points then an optimal (k-1)-level quantizer can always be improved by adding in one more quantization level".
> > > >
> > > > Especially, when encoder $f_e$ is modeled by a neural network that makes the model easily stuck in a local minimum. Therefore, we believe regularization is necessary. Additionally, due to the codebook being learnable, in the worst case, some centroids might converge to the same value, hence increasing size for specific centroids.

---

> > ### Comment · Reviewer_9V99 · 2022-11-18
> > **Theorem 3.1**
> >
> > I strongly disagree that Theorem 3.1 is novel. A $K$-level vector quantizer $q$ is usually decomposed into an encoder $\alpha: \mathcal{X} \to [K]$ where $[K]=\{1, \dotsc, K\}$ and decoder $\beta: [K] \to \mathcal{X}$ so that $q = \beta\circ\alpha$ [Gersho and Gray, 1992]. Then Pollard's result is that $\min_{\alpha,\beta}\mathbb{E}d(X, \beta\circ\alpha(X)) = W_d( P_X, q{\sharp} P_X)$, and $q\sharp P_X = \sum_{i=1}^K \pi_i \delta_{x_i}$ for some $\pi_i$ and $x_i$. So Pollard's result is equivalent to Theorem 3.1 if $\mathcal{Z}=[K]$. Now suppose $\mathcal{Z}\subset\mathbb{R}^D$. There always is a one-to-one mapping $f$ between $[K]$ and {$c_1, \dotsc, c_K$} $\subset \mathcal{Z}$. So if $f_e:\mathcal{X}\to\mathcal{Z}$ is the optimal encoder and $f_d:\mathcal{Z}\to\mathcal{X}$ is the optimal decoder from Theorem 3.1, then one can always write $f_e = f\circ\alpha$ and $f_d = \beta\circ f^{-1}$ and vice versa.
> >
> >
> >
> > ### Reference
> >
> > Gersho, A. and Gray, R.M., 1992. Vector quantization and signal compression. Kluwer Academic Publishers, Boston.

---

> > > ### Author Response · Authors · 2022-12-02
> > > **Response to Reviewer 9V99 (Theorem 3.1)**
> > >
> > > Thanks for giving us your feedback about Theorem 3.1.
> > >
> > > We are pleasantly surprised to learn of the connections between Theorem 3.1 and the results of Pollard (1982).
> > >
> > > We consider the in the result of pollard $\\min\_{\\alpha, \\beta}\\mathbb{E}[d(X, \\beta \\circ \\alpha(X))] = \\mathcal{W}\_d(P\_X, q\\# P(X))$. Suppose $\mathcal{Z} \in \mathbb{R}_D$. There always is a one-to-one mapping $f$ between $[K]$ and $\{c_1, c_2, ..., c_K\} \in \mathcal{Z}$ . We can write $f_e=f \circ \alpha$ and $f_d = \beta \circ f^{-1}$. The result of Pollard becomes:
> > >
> > > - $$
> > > \\min\_{f\_e, f\_d}\\mathbb{E}[d(X, f\_d \\circ f\_e(X))] = \\mathcal{W}\_d(P\_X, q\\# P(X))
> > > $$
> > > where $q=(f\circ \alpha) \circ (\beta \circ f^{-1})$.
> > >
> > >
> > > After inspecting, we observe that, in the result of pollard, $C$ actually is fixed i.e.,  $P\_{c,\\pi}= (f \\circ \\alpha)\\# P(X)$ depends on $q$. However, $C$ is trained in our formulation (theorem 3.1 and OP (1)):
> > >
> > > - $$
> > > \\min\_{C,\\pi,f\_d} \\min\_{\\bar{f}\_{e} : \\bar{f}\_{e} \\# \\mathbb{P}\_{x}=\\mathbb{P}\_{c,\\pi}} \\mathbb{E}\_{x\\sim\\mathbb{P}\_{x}}\\left[d\_{x}\\left(f\_{d}\\left(\\bar{f}\_{e}\\left(x\\right)\\right),x\\right)\\right] =
> > > \\min\_{C,\\pi}\\min\_{f^{d}}\\mathcal{W}\_{d\_{x}}\\left(f\_{d}\\#\\mathbb{P}\_{c,\\pi},\\mathbb{P}\_{x}\\right)
> > > $$
> > >
> > > The optimal transport value of two results (ours and pollard) are equal but the solutions of $C$ are not necessarily the same as in our formulation $C$ can be arbitrary and $f_e$ and $f_d$ should fit with $C$ to obtain optimal $\mathcal{W}_d$. In other words, two results are only equivalent when $C$ is fixed in our formulation.
> > > However, we also would like to admit that the results of Pollard can be extend to our theorem 3.1 with non-trivial derivations (i.e., the encoder $f_e: \mathcal{X} \rightarrow \mathcal{X}$ in general maps data to continuous latent $\mathcal{Z}$ which is unable to simply assume $\mathcal{Z} = [K]$).
> > >
> > > It's worth to noting that $C$ being trainable is  important in practical algorithm (ref. our theorem 3.2) as we optimize the distance on latent space $\mathcal{Z}$ instead data space $\mathcal{X}$ (theorem 3.1 and pollard's result measure the distance on data space). $C$ being trainable will help train $f_e$ and $f_d$ easier. We will clarify it in final version. Despite the similarity, we believe our derivation provides additional motivation for the technique of auto-encoder in form of neural network.
> > >
> > > Additionally, our novelties is not only from Theorem 3.1 but also our theoretical pathway, particularly Theorem 3.2 and Corollary 3.1 which lead to our practical algorithm. We appreciate the reviewer if could reconsider these aspect on out novelties.

---

### Decision · Program_Chairs · 2023-01-20

**Decision:**

Reject

**Justification For Why Not Higher Score:**

requires a major revision

**Justification For Why Not Lower Score:**

N/A

**Metareview: Summary, Strengths And Weaknesses:**

### Description

The goal of the work is to learn discrete latent representations and a pair of deterministic mappings between these latent representations and space of observations. The encoder mapping induces a clustering in the space of observations. The decoder mapping induces etalons in the space of observations. The objective is to minimize the reconstruction error (1) where only the decoder matter. The paper proposes a new theoretically derived approach (and some approximation steps) to solve the problem by employing also a deterministic encoder into the vector space of codebook vectors. This design is motivated by VQ-VAE (Van Den Oord et al., 2017), however the theory builds a different pass.

### Evaluation

Reviewers have put forward serious arguments for the novelty and clarity of the theoretical derivation and the gap between the theoretical derivation and the actual algorithm implemented. In particular reviewers 9V99 and 7LM4 pointed many substantial technical concerns.
The discussion did not fully converge argument-wise and the amount of revision work to the technical core of the paper appears to be exceeding what can be handled within the review. Reviewers did not change their scores. In particular in the concluding comment reviewer 7LM4 says that the correction of the first part of Appendix A is ok, but the argumentation from the perspective of the practical implementation was not persuasive.

I believe the authors should revise the paper and carefully build their line of arguments on all of the debated questions.

### Additional feedback

Sorry, I cannot follow all the technical contribution, but have a few comments which might help improve the presentation

1) I believe that $d_z (sg(f_e (x)),C) + \beta d_z (f_e (x),sg(C))$
is equivalent to just $d_z (f_e (x),C)$ with the learning rate for encoder parameters adjusted by $\beta$. It should be defined that it is to be minimized. IS the objective an upper bound on $-\log p(x)$ as one would expect for a VAE model? Furthermore, it is not clear what is the function of the $d_z (f_e (x),C)$ loss (total distance from encoder vector to all centroids). The optimum wrt to centroids $C$, as $d_z (f_e (x),C)$ is separable over $c$ as defined, assuming d_z is Euclidean, is the mean embedding $E[f_e(X)]$, same for all centroids, which makes little sense.

2) The second concern I have, is that the introduction of the codebook is not sufficiently justified.
A general neural network decoder may have (and likely has in practice) a layer $R^K \to R^D$, which can implement an arbitrary embedding, equivalently representing a learnable codebook. Thus a fixed codebook, \eg with basis vectors would suffice from the modelling perspective. What is the advantage of having the codebook explicitly in the problem formulation? Does it allow for a more efficient optimization scheme, or is necessary for a particular theoretical derivation step later on?

3) I could not catch up on the technical arguments about the question of how the theory extends to the case of $M \times M$  codebooks. From what I can see, this latent space corresponds to a $K^{M \times M}$ mixture components in $P_{c,\pi}$, leading to a high complexity (intractability) of some constructs (ie. the assignment function $\sigma$, the encoder $\bar f_e$) and potentially affecting all proofs (ie. Lemma 3.1 and on) and widens the gap between theory and practice. In particular, the assumption that a general $\bar f_e$, which is a discrete encoder in the product space, can be realistically modelled by $f_e : X \to R^{K M^2}$ in Theorem 3.2 appears not plausible.

4) The straight-through heuristics for training the discrete argmin encoder, inherited from the prior, work does not appear sound.



**Summary Of Ac-Reviewer Meeting:**

N/A